# Nestin-dependent mitochondria-ER contacts define stem Leydig cell differentiation to attenuate male reproductive ageing

Senyu Yao[1,2,8], Xiaoyue Wei[1,2,8], Wenrui Deng[1,2,8], Boyan Wang[1,2,8], Jianye Cai [2,3], Yinong Huang[2,4], Xiaofan Lai[2,5], Yuan Qiu[2], Yi Wang [2], Yuanjun Guan[6] & Jiancheng Wang [1,2,7✉]

Male reproductive system ageing is closely associated with deficiency in testosterone production due to loss of functional Leydig cells, which are differentiated from stem Leydig cells (SLCs). However, the relationship between SLC differentiation and ageing remains unknown. In addition, active lipid metabolism during SLC differentiation in the reproductive system requires transportation and processing of substrates among multiple organelles, e.g., mitochondria and endoplasmic reticulum (ER), highlighting the importance of interorganelle contact. Here, we show that SLC differentiation potential declines with disordered intracellular homeostasis during SLC senescence. Mechanistically, loss of the intermediate filament Nestin results in lower differentiation capacity by separating mitochondria-ER contacts (MERCs) during SLC senescence. Furthermore, pharmacological intervention by melatonin restores Nestin-dependent MERCs, reverses SLC differentiation capacity and alleviates male reproductive system ageing. These findings not only explain SLC senescence from a cytoskeleton-dependent MERCs regulation mechanism, but also suggest a promising therapy targeting SLC differentiation for age-related reproductive system diseases.

[1] Scientific Research Center, The Seventh Affiliated Hospital of Sun Yat-sen University, Shenzhen 518107, China. [2] Center for Stem Cell Biology and Tissue Engineering, Key Laboratory for Stem Cells and Tissue Engineering, Ministry of Education, Sun Yat-Sen University, Guangzhou 510080, China. [3] Department of Hepatic Surgery and Liver Transplantation Center of the Third Affiliated Hospital, Organ Transplantation Institute, Sun Yat-Sen University, Guangzhou 510630, China. [4] Department of Endocrinology, The First Affiliated Hospital of Sun Yat-Sen University, Guangzhou 510080, China. [5] Department of Anesthesiology, The First Affiliated Hospital, Sun Yat-sen University, Guangzhou 510080, China. [6] Core Facility of Center, Zhongshan School of Medicine, Sun Yat-Sen University, Guangzhou 510080, China. [7] Department of Hematology, The Seventh Affiliated Hospital, Sun Yat-Sen University, Shenzhen 518107, China. [8] These authors contributed equally: Senyu Yao, Xiaoyue Wei, Wenrui Deng, Boyan Wang. ✉email: wangjch38@mail.sysu.edu.cn

The reproductive system is of crucial importance to many processes related to the development, maturation and ageing of males[1]. It has been reported that reproductive system ageing implicates other organs, such as the heart and bones, which may lead to overall senescence of the whole organism[2,3]. Testosterone is the major anabolic hormone in men and a key player in maintaining normal reproductive and sexual function. However, the population mean of serum testosterone level declines with age, leading to ageing of the reproductive system. Therefore, it has been suggested that testosterone deficiency may be the driving force that propels the overall senescence of males[4].

Leydig cells (LCs) account for most testosterone production in men[5]. During ageing, the testosterone-synthesizing capacity in LCs gradually decreases, which is considered the major reason for the age-related drop in testosterone levels[6]. Although numerous studies have explored approaches to restore testosterone production in aged LCs[7,8], few of them have yielded results with clinical significance, indicating the complexity of regulating the cellular state of LCs. However, it is still feasible to restore testosterone production by reinforcement of newly differentiated LCs. Stem Leydig cells (SLCs), which are mesenchymal stem cells (MSCs), reside in the testicular interstitium. When cultured in vitro, SLCs can be induced to differentiate into LCs with normal testosterone-producing capacity[9]. In addition, in living organisms, SLCs are capable of proliferating and differentiating into functional LCs, but SLCs are likely to be damaged under pathological conditions or senescence[10]. These findings have led us to hypothesize that during ageing, the loss of differentiation capacity in SLCs may account for the reduction in the healthy LC population. By uncovering the mechanism of SLC senescence, we may be able to restore the differentiation capacity of aged SLCs, thus re-establishing a functional LC population and ultimately, alleviating the ageing process.

The mechanism of stem cell ageing is heterogeneous. Segel et al. identified the mechanoresponsive ion channel Piezo1 as a key mediator of oligodendrocyte progenitor cell ageing, which corresponds to the fact that the prefrontal cortex progressively stiffens during senescence[11]. For bone marrow-derived mesenchymal stem cells, their ageing is regulated by autophagy levels, and inhibiting autophagy could turn young cells into an aged state with degenerative properties[12]. However, the ageing mechanism of various kinds of stem cells, such as SLCs in the reproductive system, remains unknown. Multiple organs in the reproductive system display particularly high steroidogenesis activity due to the production of sex hormones. In the testis, the production of testosterone by LCs requires the delicate cooperation of enzymes that are located on different organelles, especially mitochondria and the endoplasmic reticulum (ER). The multistep process that converts cholesterol into the final steroid hormone product involves the transportation of testosterone precursors through close contact sites between these two organelles, namely, mitochondria-ER contacts (MERCs)[6]. Considering its central role in controlling steroidogenesis, MERCs should have particular significance in the reproductive system. Moreover, Ziegler et al. demonstrated that MERCs also participate in the senescence of human cells, suggesting that in stem cells, the age-related decline in differentiation capacity may be associated with dysfunctional MERCs[13]. Together, these findings suggest that maintaining normal MERCs could be an ideal approach to preserve the differentiation capacity of SLCs, thus mitigating reproductive system senescence.

Nestin is a type VI intermediate filament (IF) protein originally identified as a marker for neural stem cells in early development[14]. It was later found to be expressed in many other stem/progenitor cell populations, promoting its application as a marker to isolate various adult stem cells, including stem Leydig cells[15]. As an intermediate filament, Nestin is highly dynamic in expression level, which ensures its ability to mediate intracellular activity in response to various stimulations. In recent years, emerging studies have further revealed Nestin's function as an important regulator, altering various cellular activities, especially in proliferative cells, which seems distinctive among IFs. In our previous work, we demonstrated that Nestin colocalizes with mitochondria, and that knockdown of Nestin influences the mitochondrial function, including oxygen consumption rates, ATP generation and mitochondrial membrane potential[16]. Moreover, Nestin has also been found to participate in the maintenance of stemness in neural stem/progenitor cells[17]. Therefore, uncovering the function of Nestin in regulating intracellular homeostasis and the differentiation capacity of stem Leydig cells will provide new insights into therapies for male age-related reproductive diseases.

Here, we report that SLCs from aged mice exhibit lower differentiation capacity. It is postulated that this reduction in stemness is due to impaired MERCs, which in turn results in imbalanced intracellular homeostasis, including disordered redox homeostasis and increased ER stress. We further identifies Nestin as an important regulator of MERCs, which reacts upon elevated ROS stress by competing with Nrf2 in binding Keap1. Intriguingly, melatonin treatment significantly rescues MERCs deficiency in aged SLCs, thus restoring their differentiation capacity. Moreover, in vivo study shows that improving MERCs are effective in improving overall male function in aged mice, providing a promising path to stem cell therapy in reproductive system ageing.

## Results

**SLCs exhibit lower capacity for differentiation during ageing.** To determine the exact reason for the age-related decline in testosterone levels, we investigated changes that occurred in the testosterone-producing cell population during ageing. H&E staining of mouse testicular mesenchyme revealed a significant decrease in interstitial cell number during ageing, mostly resulting from the reduction of LCs (Fig. 1a). Along with the reduction in serum testosterone levels (Fig. 1b), lowered mRNA expression of LC markers, such as HSD3β, CYP11A1, and HSD17β3, was also noticed (Fig. 1c–e). In addition, the density of the cells expressing these markers also decreased in aged testes (Fig. 1f–i). The aforementioned result confirmed the reduction in the LC population as the major cause of the age-dependent decline in testosterone production.

SLCs are capable of proliferating and differentiating into functional LCs, playing an important role in maintaining the number of LCs and intratesticular environmental homeostasis[18]. Thus, to determine whether the shrinkage in LC colonies is caused by dysfunction in generating new LCs, we tracked the cell fate of Nestin-expressing SLCs by labeling them with RFP (Fig. 1j). Interestingly, we found that there was a prominent reduction in newly differentiated LCs, shown as RFP + LHR + cells, relative to the total number of RFP + cells in old mice, suggesting that the impaired differentiation of SLCs into LCs chiefly contributed to the decreasing number of LCs in senescent testes (Fig. 1k–l).

To further confirm the effect of ageing on SLC differentiation, we harvested Nestin+ SLCs from mice of different ages and induced them to differentiate in vitro (Supplementary Fig. 1a). Consistent with previous experiments, we identified that SLCs from old mice produced fewer mature LCs (Supplementary Fig. 1b–j). Together, these results indicate that the age-dependent decrease in the differentiation capacity of SLCs leads to a reduced LC population, thus resulting in insufficient testosterone production during ageing.

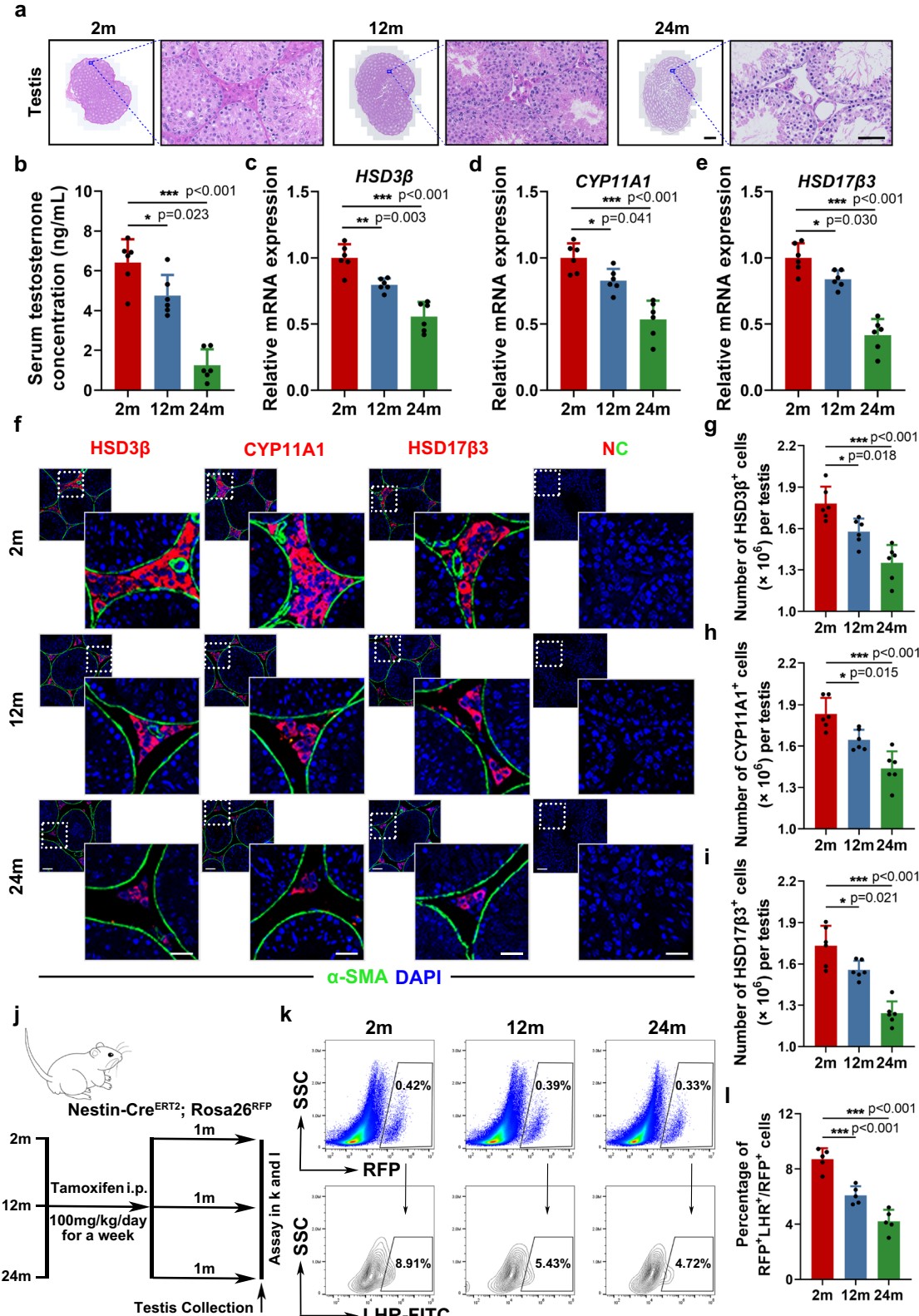

**SLCs establish fewer MERCs during ageing.** The mitochondria-ER contacts (MERCs) are crucial for carrying out various cellular functions[19], including mitochondrial dynamics, Ca$^{2+}$ signaling and lipogenesis. Considering the active lipid metabolism in the reproductive system[5,20,21], we speculated that MERCs are particularly important in regulating the differentiation of SLCs into

LCs. Therefore, we further investigated whether the loss of differentiation capacity of aged SLCs results from the decoupling between mitochondria and the ER. Colocalization analysis revealed that mitochondria and ER displayed less contact during ageing (Fig. 2a–d). Meanwhile, at the ultrastructural level, we defined "effective apposition" as sites where the distance between

**Fig. 1 SLCs exhibit lower capacity for differentiation into LCs during ageing. a** Representative H&E staining pictures of testicular mesenchyme from different age groups (2 months, 12 months, and 24 months). Scale bar, 500 μm for original pictures and 50 μm for enlarged pictures. **b** Serum testosterone level of different age groups. (*n* = 6 biological repeats for each group; One-way ANOVA). **c–e** qPCR analysis of relative mRNA expression of LCs markers of testes from different age groups. (*n* = 6 biological repeats for each group; One-way ANOVA). **f** Representative immunostaining pictures of testicular mesenchyme from different age groups. LCs are identified as HSD3β + /CYP11A1 + /HSD17β3 + cells. Scale bar, 40 μm for original pictures and 20 μm for enlarged pictures. **g–i** Quantification of LCs cell numbers with different markers in (**f**). (*n* = 6 biological repeats for each group; One-way ANOVA). **j** Schematic of the experimental animal model for labeling and tracing Nestin+ cells. **k** Flow cytometry for detecting the percentage of RFP + LHR + cells in total RFP + cells of different age Nestin-CreERT2; Rosa26RFP mice. **l** Quantification of percentage of RFP + LHR + cells in total RFP + cells of different age Nestin-CreERT2; Rosa26RFP mice in (**k**) (*n* = 5 biological repeats for each group; One-way ANOVA). Two-sided comparison; All data are mean ± SD; Error bars represent SDs. *$p < 0.05$, **$p < 0.01$, ***$p < 0.001$; Source data are provided as a Source Data file. See also Supplementary Fig. 1.

mitochondria and the ER was less than 100 nm[22] and we found that fewer effective apposition sites could be observed in SLCs from older mice, and a smaller proportion of mitochondria was in close contact with the ER (Fig. 2e–g and Supplementary Fig. 2a, b). This phenomenon was robust, as we repeated our quantification using a variety of interorganellar tethering lengths—mito-ER distances of 100, 50, and 25 nm (Supplementary Fig. 2c–f), which all pointed to the same conclusion. Moreover, we applied surface-surface contact site area assay and proximal ligation assay (PLA) to further illustrate the age-related decline in MERCs, which were also consistent with our previous results, indicating fewer MERCs (Fig. 2h–j and Supplementary Fig. 2g–j). Together, these results suggest that senescence may exert its influence on SLCs by scissoring mitochondria and ER apart.

**Fewer MERCs result in mitochondria and ER dysfunction.** Intracellular homeostasis is largely dependent on the normal function of mitochondria and the ER[23,24]. The disruption of MERCs usually leads to impaired intracellular homeostasis, mainly characterized by an imbalanced oxidative state and increased ER stress[25,26]. First, we observed that there was an increase in intracellular ROS levels, especially mitochondrial ROS, and a reduced antioxidation ability in SLCs from aged mice (Fig. 3a–e and Supplementary Fig. 3a–m). In addition, mitochondria became smaller during SLC senescence (Fig. 3f–i and Supplementary Fig. 4a, b). Moreover, higher apoptosis, lower mitochondrial membrane potential and enhanced mitophagy but lower autophagy as SLCs aged were possibly relevant to the increased ROS level (Supplementary Fig. 4c–n). Regarding mitochondrial functions, there was an obvious decline in ATP production. (Supplementary Fig. 4o). Besides, reduced abilities of glucose uptake and lactate biosynthesis were found during SLC ageing, indicating a decline in glycolysis. (Supplementary Fig. 4p, q). In addition, by analysing the expression of classical ER stress markers, we identified deteriorated ER homeostasis in aged SLCs (Fig. 3j–o). In addition, we detected changes in mitochondria-associated ER membrane (MAM) proteins, which play an important role in mitochondrial dysfunction and ER stress[27,28], and we found compensatory upregulation of these proteins as the SLCs aged, indicating that they might be downstream of the reduction in MERCs (Supplementary Fig. 5a–c). Furthermore, we also found reduced levels of antioxidation genes and elevated levels of ER stress genes in SLCs when the MAM proteins were knockdown, indicating these MAM proteins were indeed crucial to play important biological functions (Supplementary Fig. 5d–w). Altogether, these data demonstrated that intracellular homeostasis of SLCs is under delicate regulation of MERCs, which is gradually impaired during ageing.

**Keap1-mediated Nestin degradation separates MERCs.** The cytoskeleton is crucial for the communication between organelles[29,30]. Consequently, we investigated the quantitative and

morphological changes in three major types of cytoskeletons and found that intermediate filament Nestin expression sharply decreased with a more uneven distribution in aged SLCs, yet little change was present in microtubules and microfilaments (Fig. 4a–h and Supplementary Fig. 6a–d), indicating that the decline in normally distributed Nestin, which was considered as the major expression of intermediate filaments in SLCs[15], may play a crucial role in the impairment of intracellular homeostasis in SLCs.

Furthermore, analysis of Nestin degradation indicated that the age-related loss of Nestin resulted from increased ubiquitination via the ubiquitin–proteasome pathway (Fig. 4i–k). However, the mechanism of the increased ubiquitination of Nestin remains unaddressed. Kelch-like ECH-associated protein 1-nuclear factor-E2-related Factor 2 (Keap1-Nrf2) is a renowned antioxidation pathway that controls the intracellular redox state. Under normal conditions, Keap1 degrades Nrf2 through rapid ubiquitination. When the cell is threatened by elevated oxidative stress, however, Keap1 undergoes modification and loses its ability to ubiquitinate Nrf2, allowing it to accumulate in the nucleus to induce the expression of its target genes[31]. Moreover, Keap1 was found to be able to bind Nestin, which is competitive with the binding between Keap1 and Nrf2[32]. After Keap1 knockdown, we found that the ubiquitination of Nestin decreased, indicating that Keap1 is essential in mediating the ubiquitin-proteasomal degradation of Nestin (Fig. 4l). There was also a gradual shift in the binding target of Keap1: Keap1 preferentially bound to Nestin, while Nrf2-Keap1 binding was reduced in aged SLCs (Fig. 4m, n).

In addition, in order to further illustrate the functional significance of Nestin in MERCs, we used two Nestin-short hairpin RNAs (ShNES-1 or ShNES-2) as previously reported to specifically reduce Nestin expression in SLCs[33,34] (Supplementary Fig. 7a–c). As we expected, after knocking down Nestin, the level of MERCs proximity was reduced (Supplementary Fig. 7d–f). Moreover, SLCs underwent senescence as evidenced by SA-β-Gal staining, along with increased intracellular ROS levels and decreased differentiation capacity (Supplementary Fig. 7g–n). Based on the aforementioned results, it can be concluded that Nestin insufficiency leads to dysfunctional MERCs via the Keap1-Nestin/Nrf2 competitive binding mode, thus impairing SLC intracellular homeostasis and differentiation capacity into LCs.

**Targeting Nestin degradation restores aged SLCs homeostasis.** Melatonin, an endogenous hormone synthesized by the pineal gland, has been proved to be downregulated in aged organism[35,36], and could have antioxidant capacity in a variety of ways to reduce elevated ROS levels during ageing[37–39]. In addition, melatonin is capable of protecting mitochondria[40], relieving ER stress[41], and even enhancing Nrf2 expression while inhibiting Keap1[42], besides it has been safely used clinically[43,44]. Given the above reasons, we finally chose melatonin for treatment and further explored its potential for restoring SLC functions. We found that melatonin successfully restored Nestin protein levels

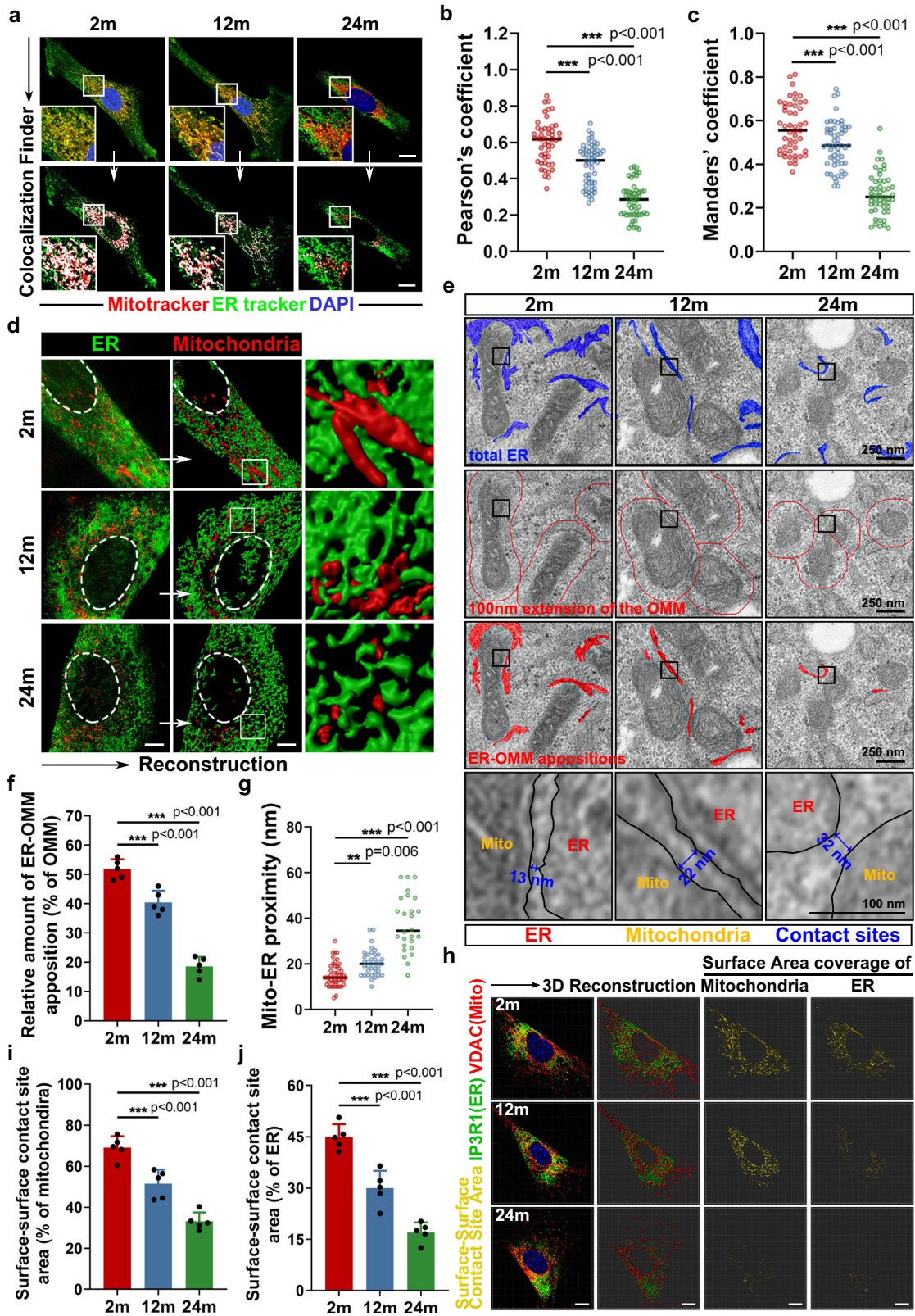

in aged SLCs in vivo (Fig. 5a), which resulted from inhibited ubiquitin-proteasomal degradation (Fig. 5b). As we expected, melatonin treatment in vivo could also restore MERCs in aged SLCs, providing stronger proof of the relationship between Nestin and MERCs (Fig. 5c). Furthermore, in addition to decreased ROS stress, restored redox homeostasis and alleviated ER stress were observed in SLCs after melatonin treatment, and mitochondrial dysfunction was also prevented (Fig. 5d–f, Supplementary Fig. 8a–o and Supplementary Fig. 9a–h). Moreover, the expression of numerous LC markers and testosterone production were both rescued after melatonin treatment (Fig. 5g and Supplementary Fig. 10a–h). In addition, melatonin's efficacy was

**Fig. 2 SLCs establish fewer mitochondria-endoplasmic reticulum contacts during ageing. a** Representative immunostaining pictures of colocalization between mitochondria and ER in primary Nestin-GFP + SLCs from different age groups. Mitochondria and ER are marked with Mitotracker and ER tracker, respectively. Scale bar, 10 μm. **b, c** Quantification of the levels of colocalization in (**a**) (Manders' (of mitochondria) and Pearson's coefficients are shown for each condition, $n = 49$, 55 and 51 cells for 2 m, 12 m, and 24 m SLCs, respectively; One-way ANOVA). **d** Representative immunostaining pictures and 3D reconstruction of mitochondria and ER in primary Nestin-GFP + SLCs from different age groups. Scale bar, 5 μm. **e** Representative transmission electron microscope (TEM) images of primary Nestin-GFP + SLCs from different age groups. Scale bar, 250 nm for original pictures and 100 nm for enlarged pictures. **f** Quantification of the extent of MERCs (<100 nm) from (**e**) in primary Nestin-GFP + SLCs from different age groups. The relative amount of ER-OMM apposition in percentage of OMM refers to the interaction length of ER tubules within 100 nm of the OMM covered in the percentage of total OMM per mitochondria ($n = 26$ to 48 mitochondria in 5 fields per condition; One-way ANOVA). **g** Quantification of the mito-ER proximity from **e** in primary Nestin-GFP + SLCs from different age groups ($n = 26$ to 48 mitochondria in 5 fields per condition; One-way ANOVA). **h** Representative immunostaining pictures, 3D reconstruction and surface-surface contact site area coverage of VDAC (mitochondria) or IP3R1(ER) in primary Nestin-GFP + SLCs from different age groups. Scale bar, 10 μm. **i** Quantification of the surface-surface contact site area (percentage of mitochondria) in (**h**) ($n = 5$ biological repeats for each group; One-way ANOVA). **j** Quantification of surface-surface contact site area (percentage of ER) in (**h**) ($n = 5$ biological repeats for each group; One-way ANOVA). Two-sided comparison; All data are mean ± SD; Error bars represent SDs. **$p < 0.01$, ***$p < 0.001$; Source data are provided as a Source Data file. See also Supplementary Fig. 2.

investigated ex vivo (Fig. 5h). Nestin expression was also rescued in the aged group by melatonin treatment (Supplementary Fig. 10i, j), followed by increased HSD3β expression and testosterone production (Fig. 5i–k). In addition, the same phenomena such as increased MERCs and enhanced differentiation capacity in aged SLCs were found in other classic antioxidants, such as NAC and VitE[45,46] (Supplementary Fig. 11a–h). Furthermore, we confirmed the efficacy of melatonin in a senescence-inducing model by applying $H_2O_2$ to young SLCs (Supplementary Fig. 12a–o). Finally, the functional link between melatonin and MERCs was further verified by loss-of-function experiments (Supplementary Fig. 13a–g). Altogether, these multimodel data repeatedly draw a clear association between melatonin availability and MERCs formation through Nestin.

**Melatonin promotes male reproductive functions in old mice.** We next applied melatonin treatment to mice to combat the age-related decline in numerous male functions, such as testosterone production and spermatogenesis. The results revealed that Nestin expression was successfully restored after melatonin treatment (Fig. 6a, b). In addition, melatonin treatment restored MERCs in aged SLCs in situ (Fig. 6c–e). Moreover, more mature LCs could be seen in the testicular interstitium from melatonin-treated mice, together with elevated LC markers expression and enhanced testosterone production (Fig. 6f, g and Supplementary Fig. 14a–d). Given that the action of testosterone is known to be critical for the completion of meiosis and spermatogenesis in rodents[47,48], we further tested whether melatonin could restore the seminiferous function of old mice and found that the expression of the meiotic marker SYCP3 was increased in the seminiferous tubules (Fig. 6h–j), which was also evidenced by H&E histological analysis of the testes (Supplementary Fig. 14e, f). Notably, the epididymal sperm number and sperm motility appeared to be significantly higher in the melatonin-treated aged mice (Fig. 6k and Supplementary Fig. 14g–h). In addition, we further detected the weight of organs associated with the reproductive system and found an increased testis volume and weight, accompanied by a decrease in the weight of the epididymis, seminal vesicle, and prostate present in melatonin-treated old mice (Supplementary Fig. 14i–m). Altogether, these results indicate that melatonin treatment significantly restored reproductive function in old male mice, thus showing great potential for combating various age-related male diseases.

Furthermore, we sought to confirm whether Nestin was necessary to rescue LC numbers by melatonin treatment. We specifically knocked out Nestin in SLCs by constructing PDGFRα-Cre[ERT2]; Nestin[loxP/loxP]; Rosa26[RFP] mice, which received melatonin

treatment with PDGFRα-Cre[ERT2]; Rosa26[RFP] control group was in parallel (Fig. 6l). The efficacy of Nestin knockout was confirmed by costaining Nestin with the SLC marker PDGFRα in vivo and in vitro (Fig. 6m, n). Consistent with our hypothesis, Nestin-knockout mice presented with fewer LCs in the testicular mesenchyme and lower serum testosterone levels, which could not be rescued by melatonin treatment (Fig. 6o–q). These findings confirmed that melatonin rescues the differentiation capacity of aged SLCs via a Nestin-dependent pathway.

**Nestin knockdown diminishes Melatonin's anti-senescence effect.** To further examine the role of Nestin in the pathogenesis of SLC differentiation in vivo, we constructed PDGFRα-Cre[ERT2]; Rosa26[CAG-LSL-Cas9-tdTomato] mice, delivered AAV8-Control or AAV8-Nestin sgRNA, and then treated them with melatonin 8 weeks later (Fig. 7a). By conditional knockout of Nestin expression in PDGFRα + SLCs in the testes (Supplementary Fig. 15a–f), ER and mitochondria exhibited significant detachment compared to the control group, which failed to be rescued by melatonin treatment (Fig. 7b, c). Functionally, melatonin treatment also failed to improve the differentiation capacity of SLCs into the LCs, testosterone production, and spermatogenesis in conditional Nestin knockout mice (Fig. 7d–h and Supplementary Fig. 15g, h). Moreover, disordered redox homeostasis and ER stress failed to be restored in the SLCs after melatonin treatment in vivo (Fig. 7i–j and Supplementary Fig. 15i, j).

Furthermore, we verified Nestin's pivotal role in regulating SLC differentiation towards LCs in a chemically damaged model[15,49]. Before melatonin treatment, the existing LCs in the testicular mesenchyme were eliminated by intraperitoneal injection of ethylene dimethanesulfonate (EDS) (Fig. 7k and Supplementary Fig. 15k). However, considering that this treatment resulted in the partial elimination of LCs in the adult testes in mice[50,51], we monitored the drug efficiency screened by serum testosterone after 4 days of treatment (Supplementary Table 1). After 4 weeks of melatonin treatment, immunostaining revealed that the number of LCs and testosterone levels recovered after melatonin treatment, yet the effect was much less significant in conditional Nestin knockout mice (Fig. 7l–n).

Altogether, the results indicated that melatonin treatment can effectively attenuate reproductive ageing in old mice. Importantly, Nestin plays an irreplaceable role in the process, probably by regulating MERCs (Fig. 8).

**Discussion**

In this study, we identified mitochondria-ER contacts (MERCs) deficiency as the main reason for the reduced differentiation

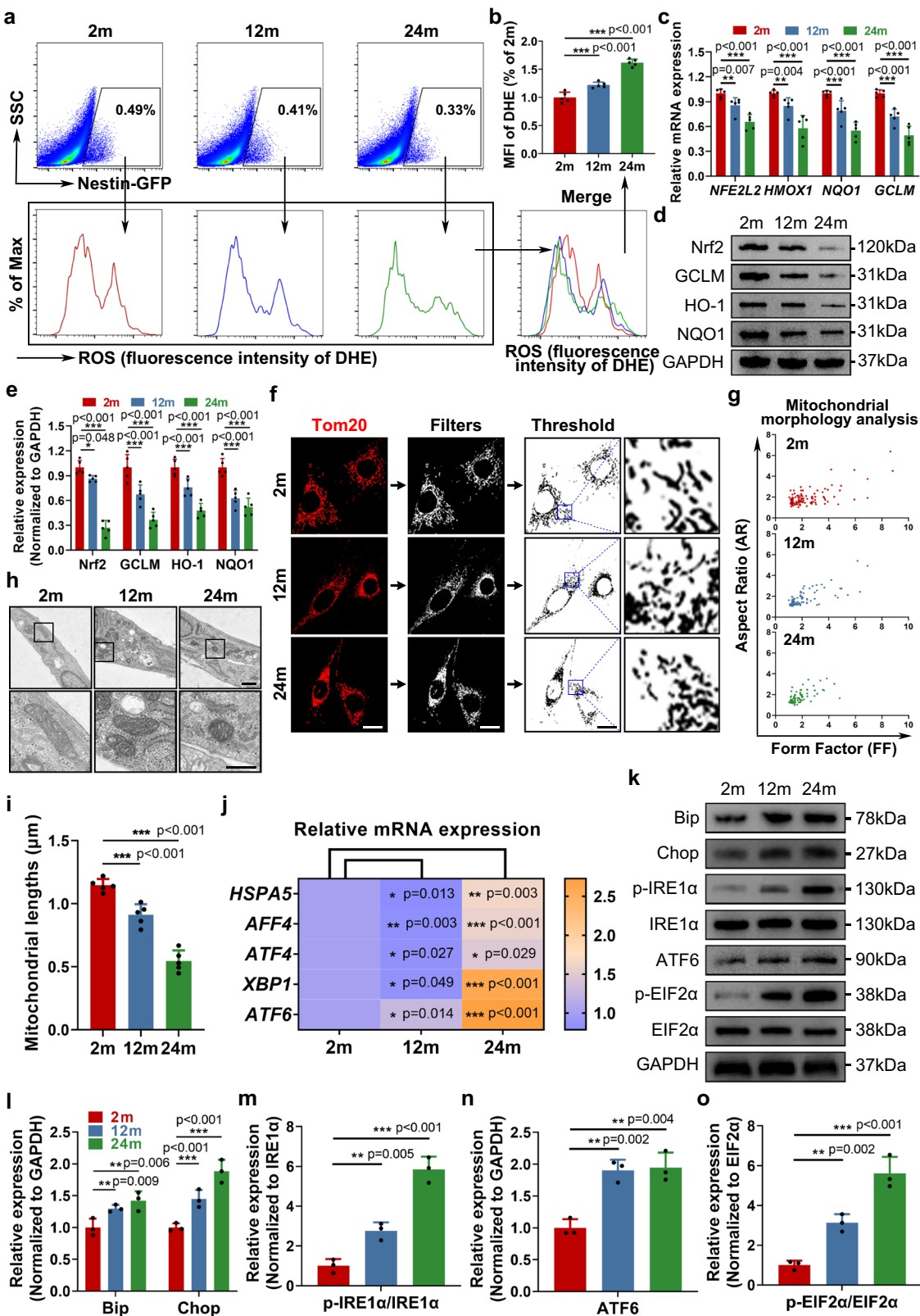

capacity in aged SLCs. This finding added to the understanding of the testosterone production drop in the aged population, apart from LC degeneration. Furthermore, we revealed a novel function of the intermediate filament Nestin in regulating stem cell differentiation, contributing to our understanding of cytoskeletal function.

Reproductive system ageing is often accompanied by decreased circulating testosterone levels. Typically, the functional decline of LCs, rather than the loss of their quantity, is considered to be the main culprit for reduced testosterone production[5]. Influenced by this theory, numerous studies have focused on the rejuvenation of aged LCs[52,53], yet none has revealed results that are promising

**Fig. 3 Fewer MERCs result in mitochondrial dysfunction and ER stress in aged SLCs. a** Flow cytometry of intracellular ROS level stained with DHE of Nestin-GFP + SLCs from different age groups in vivo. **b** Quantification of mean fluorescence intensity of DHE in (**a**). ($n = 5$ biological repeats for each group; One-way ANOVA). **c** qPCR analysis of relative mRNA expression of anti-oxidative genes of Nestin-GFP + SLCs from different age groups. ($n = 5$ biological repeats for each group; Multiple $t$ tests). **d, e** Western Blot analysis and quantification of cellular anti-oxidative protein expression of Nestin-GFP + SLCs from different age groups. ($n = 5$ biological repeats for each group; Multiple $t$ tests). **f** Representative immunostaining pictures of mitochondrial morphology of primary Nestin-GFP + SLCs from different age groups in vitro. Mitochondria are marked with Tom20. Scale bar, 20 μm. **g** A plot of Aspect Ratio (AR) against Form Factor (FF) showed that particles in primary Nestin-GFP + SLCs from different age groups in (**f**). Lower values of FF and AR, indicating small, circular mitochondria. **h** Representative TEM images of mitochondrial morphology of primary Nestin-GFP + SLCs from different age groups in vitro. Scale bar, 1 μm for original pictures and 500 nm for enlarged pictures. **i** Quantitative analysis of the average value of mitochondrial length (micrometers) in (**h**). ($n = 5$ biological repeats for each group; $n = 126–184$ mitochondria; One-way ANOVA). **j** qPCR analysis of relative mRNA expression of ER stress-related genes of primary Nestin-GFP + SLCs from different age groups in vitro. ($n = 3$ biological repeats for each group; Multiple $t$ tests). **k–o** Western Blot analysis and quantification of ER stress-related proteins of primary Nestin-GFP + SLCs from different age groups in vitro. ($n = 3$ biological repeats for each group; Multiple $t$ tests). Two-sided comparison; All data are mean ± SD; Error bars represent SDs. \*$p < 0.05$, \*\*$p < 0.01$, \*\*\*$p < 0.001$; Uncropped western blots and source data are provided as a Source Data file. See also Supplementary Figs. 3–5.

enough for clinical application. Consequently, we sought to restore testosterone production by increasing the supply of healthy, fully-functional LCs, regardless of the degenerated intrinsic LCs. By targeting MERCs in aged SLCs, enhanced differentiation towards LCs was achieved, contributing to a fortified testosterone-producing LC colony. Elevated circulating testosterone levels were observed after treatment, which indicates that the healthy LC supplied from enhanced SLC differentiation could, at least partly, offset the impact of intrinsic LC degeneration. Considering the potential risk of exogenous testosterone replacement therapy[54,55], our stem cell-targeting approach may be a promising path to recover endogenous testosterone in treating reproductive system diseases and ameliorating male ageing.

The relationship between stem cell ageing and differentiation is complicated. On the one hand, the loss of differentiation capacity is considered one of the most important signs of stem cell ageing[56]; on the other hand, excessive differentiation has been shown to deplete the stem cell pool, leading to the overall ageing of organs and systems[57]. In addition, for some types of stem cells, their ageing usually accompanies skewed differentiation into certain downstream lineages[58]. However, it remains unclear whether the senescence of stem cells damages their intrinsic differentiation modulators, or whether the extended period of differentiation causes the degeneration of stem cells. In exploring the reason for reduced testosterone production in aged mice, we solely focused on the loss of differentiation capacity of SLCs. Moreover, we observed an age-related decrease in Nestin expression levels as a potential regulator of MERCs, further influencing the differentiation of SLCs. These findings strongly suggest that the senescence of SLCs may be the root cause of reproductive system ageing, which will be elucidated deeply in future studies.

The MERCs have been proven to have great significance in maintaining lipid and calcium ($Ca^{2+}$) homeostasis, initiating autophagy and mitochondrial division, and sensing metabolic shifts[59]. More recently, Latorre-Muro et al. showed that MERCs are also required for the regulation of mitochondrial crista formation[60]. These findings imply the central role of MERCs in regulating intracellular homeostasis and maintaining the functional state of the two organelles. Considering mitochondria, an adequate level of MERCs is essential to mitochondrial health. MERCs are closely associated with mitochondrial dynamics and mitophagy, which are necessary to optimize mitochondrial quality control and maintain normal intracellular ROS levels[61]. Given the importance of ROS in regulating stem cell activity, the age-dependent deterioration of stem cell function should be closely linked to mito-ER crosstalk. Consistent with this, our findings proved that impaired MERCs in aged SLCs lead to

smaller mitochondrial morphology, reduced mitochondrial mass, ROS accumulation and decreased ATP production, which also corresponds to the reduced differentiation ability in aged SLCs. Regarding the regulation of ER, it is commonly believed that increased ER stress usually accompanies the tightening of MERCs[28], which seems contradictory to our result that impaired MERCs appear with increased ER stress simultaneously. However, impaired MERCs may serve as a perfect example of a dysfunctional regulatory system in aged stem cells. When ER stress occurs in healthy SLCs, MERCs tighten to increase ATP production, thus facilitating the synthesis of ER chaperones that contribute to the normal folding of proteins[28]. Moreover, previous studies have confirmed that MERCs are the initiation point of autophagy[62,63], which eliminates misfolded proteins and alleviates the unfolded protein response in the ER. As a result, these studies suggest that MERCs is a cellular response that will be activated transiently and alleviates ER stress when it occurs. Together, this finding further echoes the fact that MERCs play a central role in regulating the homeostasis of the reproductive system, thus controlling the differentiation of SLCs.

Various kinds of cytoskeletons have been proven to be involved in the regulation of MERCs. Friedman et al. proposed that acetylated microtubules provide tracts for mitochondria and ER transportation within the cell, thus increasing the chance for establishing contact between the two organelles[29]. Korobova et al. also suggested that actin fibers with related regulatory factors are a classical regulatory factor of MERCs[64,65]. In this study, however, no change in the expression of morphology of these two kinds of cytoskeletons was observed, suggesting that neither microtubules nor actin fibers accounted for the mito-ER detachment in aged SLCs, probably due to different cell types or different conditions. Intermediate filaments (IFs) are an enormous family of cytoskeletal proteins, comprising Lamin, Desmin, Keratin, Vimentin and Nestin, which are highly variable among different tissues and different developmental stages of a single cell. Unlike microtubules and actin fibers, the structure and function of the IF skeleton is controlled by the dynamic expression of IF proteins[66]. In addition, various studies have demonstrated that IF expression is altered as the cellular state changes[67–69], indicating that IF has regulatory functions in mediating cellular function that far exceed the basis supporting the function of the cytoskeleton. Nevertheless, due to the heterogeneity between different IF subfamilies, their functions have not been fully elucidated. Here, we reported that Nestin expression actively mediates the differentiation of SLCs through MERCs, adding to the evidence that IF proteins can regulate cellular function via expression changes, thus more widely expanding our knowledge about IF's function. Moreover, we also

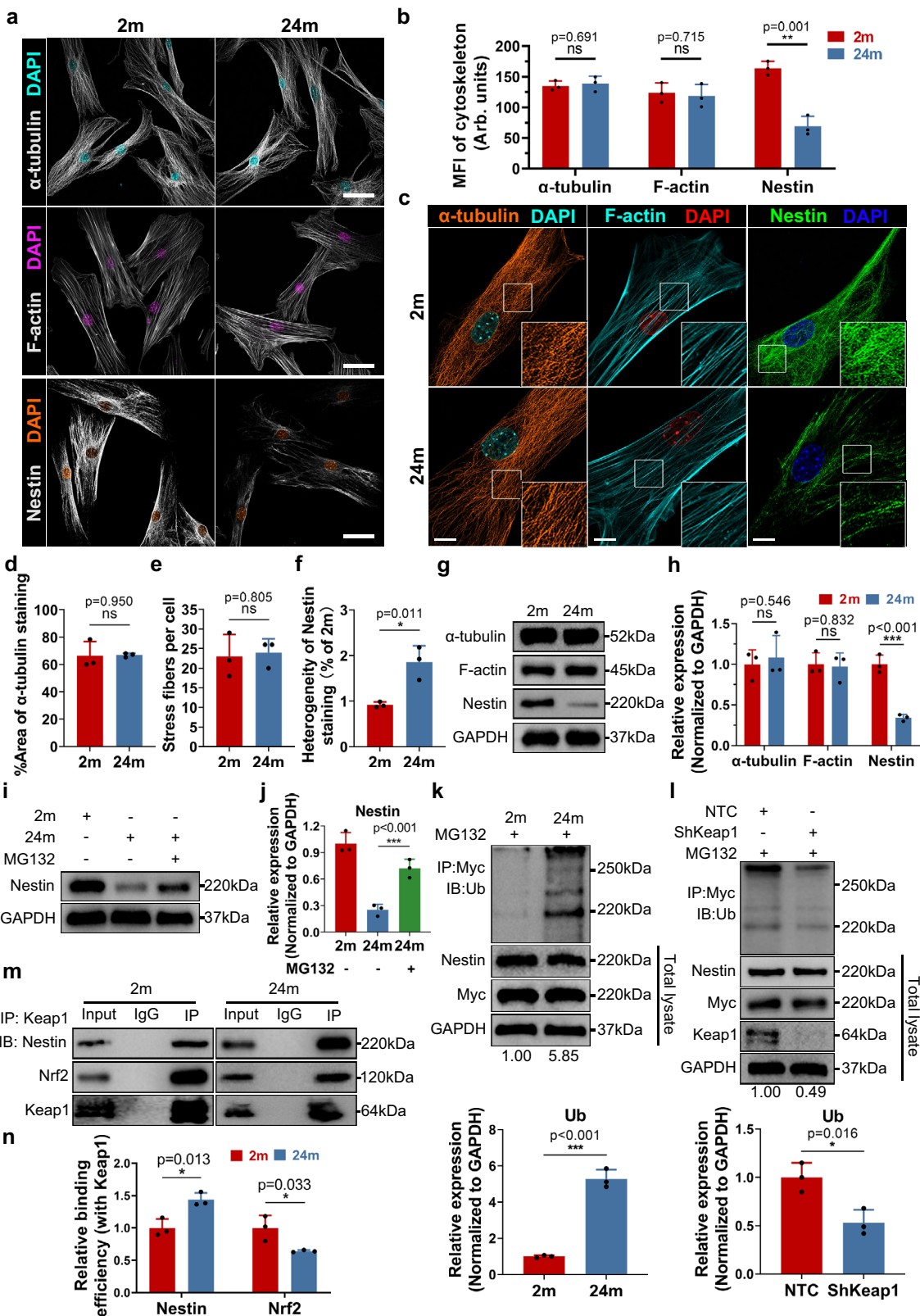

found the reduction of MERCs due to the degradation of Nestin might trigger a different compensatory change of MAM protein levels, consistent with the conclusion that the change in MAM protein levels does not always correlated with the MERCs number, indicating other molecular mechanisms regulating the number of MERCs[70–72] or the other cellular distribution and

biological function[73,74], which enlightens us to seek for deeper understanding of the relationship between Nestin and MAM proteins in the future studies. Therefore, considering Nestin's proven effect in regulating stem cell function, more Nestin-targeting therapies in rejuvenating aged stem cells are worth exploring.

**Fig. 4 Keap1-mediated Nestin degradation separates MERCs in aged SLCs. a** Representative immunostaining pictures of α-tubulin, F-actin and Nestin in primary SLCs from 2 months and 24 months old mice. Scale bar, 40 μm. **b** Quantitative analysis of the mean fluorescence intensity of three kinds of the cytoskeleton in (**a**). ($n = 3$ biological repeats for each group; Multiple $t$ tests). Arb. units, absorbance units. **c** Representative detailed immunostaining pictures of α-tubulin, F-actin and Nestin in primary SLCs. Scale bar, 10 μm. **d** Quantitative analysis of the area covered by α-tubulin in primary SLCs. ($n = 3$ biological repeats for each group; Unpaired $t$ test). **e** Quantitative analysis of the number of F-actin fibers in primary SLCs. ($n = 3$ biological repeats for each group; Unpaired $t$ test). **f** Standard deviation of the organization of Nestin fibers in primary SLCs were calculated. ($n = 3$ biological repeats for each group; Unpaired $t$ test). **g, h** Western Blot analysis and quantification of α-tubulin, F-actin and Nestin in primary SLCs. ($n = 3$ biological repeats for each group; Multiple $t$ tests). **i, j** Western Blot analysis and quantification of Nestin in primary SLCs with or without treatment with MG132. ($n = 3$ biological repeats for each group; Unpaired $t$ test). **k** Co-IP of Myc-Nestin and ubiquitin in primary SLCs after treatment with MG132. Ubiquitination level are quantified by measuring the gray-scale value of the whole lane of Western Blot band. Relative expression of ubiquitinated Nestin (normalized to GADPH) were normalized to 2 months old group. GADPH was used as loading control. ($n = 3$ biological repeats for each group; Unpaired $t$ test). **l** Co-IP of Myc-Nestin and ubiquitin in primary SLCs from 24 months old Keap1-knockdown mice after treatment with MG132. Quantification of ubiquitination level, normalization of ubiquitinated Nestin expression and loading control are indicated in (**k**). ($n = 3$ biological repeats for each group; Unpaired $t$ test).
**m, n** Immunoprecipitation analysis and quantification of Keap1–Nestin binding and Keap1-Nrf2 binding degree in primary SLCs. ($n = 3$ biological repeats for each group; Multiple $t$ tests) Two-sided comparison; All data are mean ± SD; Error bars represent SDs. *$p < 0.05$, **$p < 0.01$, ***$p < 0.001$, ns, $p > 0.05$; Uncropped western blots and source data are provided as a Source Data file. See also Supplementary Figs. 6 and 7.

In various organs and systems, the senescence of stem cells often accompanies elevated intracellular ROS levels[75,76]. Interestingly, melatonin, the indoleamine produced by the pineal gland, has been well stated for its antioxidation properties. It has been reported that melatonin can protect stem cells by scavenging radicals, upregulating antioxidant enzyme expression, reducing the activity of pro-oxidant enzymes, and maintaining mitochondrial homeostasis[77]. More encouragingly, data from animal and human studies have proven that short-term use of melatonin elicits only mild adverse effects, such as dizziness, headache, nausea, and sleepiness, even in extreme doses[78]. Here, we explored melatonin's effect in restoring Nestin expression and MERCs, yielding satisfactory results. To date, approaches to regulate intracellular Nestin levels are limited to gene ablation and RNA interference[17], which greatly hinders Nestin-targeted therapy from clinical application. Intriguingly, this finding provided evidence that Nestin expression can be altered by small molecular drugs, which is far more practical than gene therapy. Considering the important function of Nestin in multiple organs and systems[17,32], there should be wider application of Nestin-targeted therapy in other degenerative diseases, which can be explored in future studies.

In conclusion, our identification of the pivotal role of Nestin-dependent MERCs in SLC senescence opens the door for further understanding of male reproductive system ageing, providing insights into therapies for age-related diseases.

## Methods

**Animals**. Animal use and all experiments involving animals were approved by the Ethical Committee of Sun Yat-sen University. Only male mice were used throughout the experiment.

C57BL/6 mice were purchased from Beijing Vital River Laboratory, aged 2 months, 12 months, and 24 months were used for establishing comparisons between different age groups.

Nestin-GFP mice (homozygous transgenic mice expressing enhanced GFP controlled by a Nestin promoter on the C57BL/6 genetic background) were kindly provided by Dr. Masahiro Yamaguchi[79,80].

Rosa26[RFP] mice (#007914) were both purchased from Jackson Lab (Bar Harbor, America).

PDGFRα-Cre[ERT2] mice[81,82] were gifts from the Institute of Biochemistry and Cell Biology, Shanghai Institutes for Biological Sciences, Chinese Academy of Sciences.

Nestin[loxP/loxP] mice were purchased from VIEWSOLID BIOTECH V (Beijing, China).

Rosa26[CAG-LSL-Cas9-tdTomato] mice (#T002249) were purchased from GemPharmatech Co., Ltd (Nanjing, China).

Cre[ERT2]-mediated recombination was induced by administrating tamoxifen (100 mg/kg, dissolved in 150 μl corn oil, daily for a week).

Melatonin (Sigma) was dissolved in ethanol and diluted in normal saline solution to a final concentration of 5% ethanol. Mice were anesthetized with isoflurane and injected intraperitoneally with melatonin (10 mg/kg/d). Drug treatments were administered every day for 30 days. Control groups were received sham injections (without melatonin).

EDS was provided by Dr Renshan Ge (Wenzhou Medical College) and was injected intraperitoneally with either a single dose of 300 mg/kg body weight in mice. Considering that this treatment resulted in the partial elimination of LCs in the adult testis in mice[15,49], we monitored the drug efficiency screened by serum testosterone within 4 days of treatment. The detailed information is described in Supplementary Table 1.

Two versions of adeno-associated virus vector serotype 8 expressing spgRNA under the control of the human CMV promoter were purchased from the Vigene Biosciences Co., Ltd (Shanghai, China). Mice were deeply anesthetized and intra-testis administered with $8 \times 10^{10}$ GC of a pseudotyped AAV8-spgRNA-GFP vector or $8 \times 10^{10}$ GC of a pseudotyped AAV6-spgRNA-Nestin vector.

All the animals were provided free access to food and water and kept in a colony room under conditions of constant temperature (25 °C), humidity (70%), and lighting (12 h light/12 h dark cycle) in the Sun Yat-sen University Animal Center.

**Cell isolation and culture**. Primary mouse stem Leydig cells isolation was performed according to a previous article[15]. The testes were dissected from Nes-GFP or C57BL/6 mice, and then we removed the tunica albuginea carefully from each testis and minced the testes into small pieces. The interstitial cells were then digested from the seminiferous tubules with 1 mg/ml collagenase type IV (Invitrogen) in Dulbecco's modified Eagle's medium (DMEM)/F12 (1:1; Gibco) at 37 °C for 15 min. The supernatant was filtered through a 45 μm strainer and collected, and the cells in the supernatant were subsequently acquired by centrifuging at 500×g for 5 min. The single cells were washed twice with PBS, resuspended in PBS, and seeded in SLC culture medium. This medium consisted of DMEM/F12 (1:1; Gibco) supplemented with 1 nM dexamethasone (Sigma), 1 ng/ml LIF (Millipore), 5 μg/L insulin-transferrin-sodium selenite (Sigma), 5% chicken embryo extract (US Biologicals), 0.1 mM β-mercaptoethanol (Invitrogen), 1% non-essential amino acids (Hyclone), 1% N2 and 2% B27 supplements (Invitrogen), 20 ng/ml basic fibroblast growth factor (Invitrogen), epidermal growth factor (PeproTech), platelet-derived growth factor (PeproTech), and oncostatin M (PeproTech). The cultures were kept at 37 °C in a humidified 5% $CO_2$ water-jacketed incubator. The medium is changed every 3 days. For SLC culture with antioxidants, 1 μM melatonin, 1 mM NAC or 50 μM Vit E is added to the culture medium, respectively.

For some experiments, bafilomycin A1 (Baf A1, 100 nM; Sigma-Aldrich) was added to assess autophagic flux according to previous articles[83]; Cells were treated with MG132 (10 μM; Sigma-Aldrich) for 4 h to block proteasomal degradation.

**RNAi transfection**. For loss of Nestin function, retrovirus vectors (pSM2) encoding Nestin shRNA were used according to previous articles[33,34]. Scramble shRNA was used as a control. All shRNAs were constructed in our laboratory. Details on the plasmids are provided in Supplemental Table 2. ShRNA transfections were performed using the MegaTran 1.0 Transfection Reagent (OriGene) according to the manufacturer's instructions. Briefly, the lentiviruses were used to infect SLCs with Polybrene (8 μg/ml) for 4 h. The original medium was replaced with a fresh medium 12 h later. Myc-tagged Nestin was constructed using Invitrogen's Gateway System. The efficiency of knockdown was assessed by Western blot. The siRNAs were purchased from Ribobio, and their encoding vectors were transfected into cells using the Lipofectamine RNAiMAX Transfection Reagent (Invitrogen).

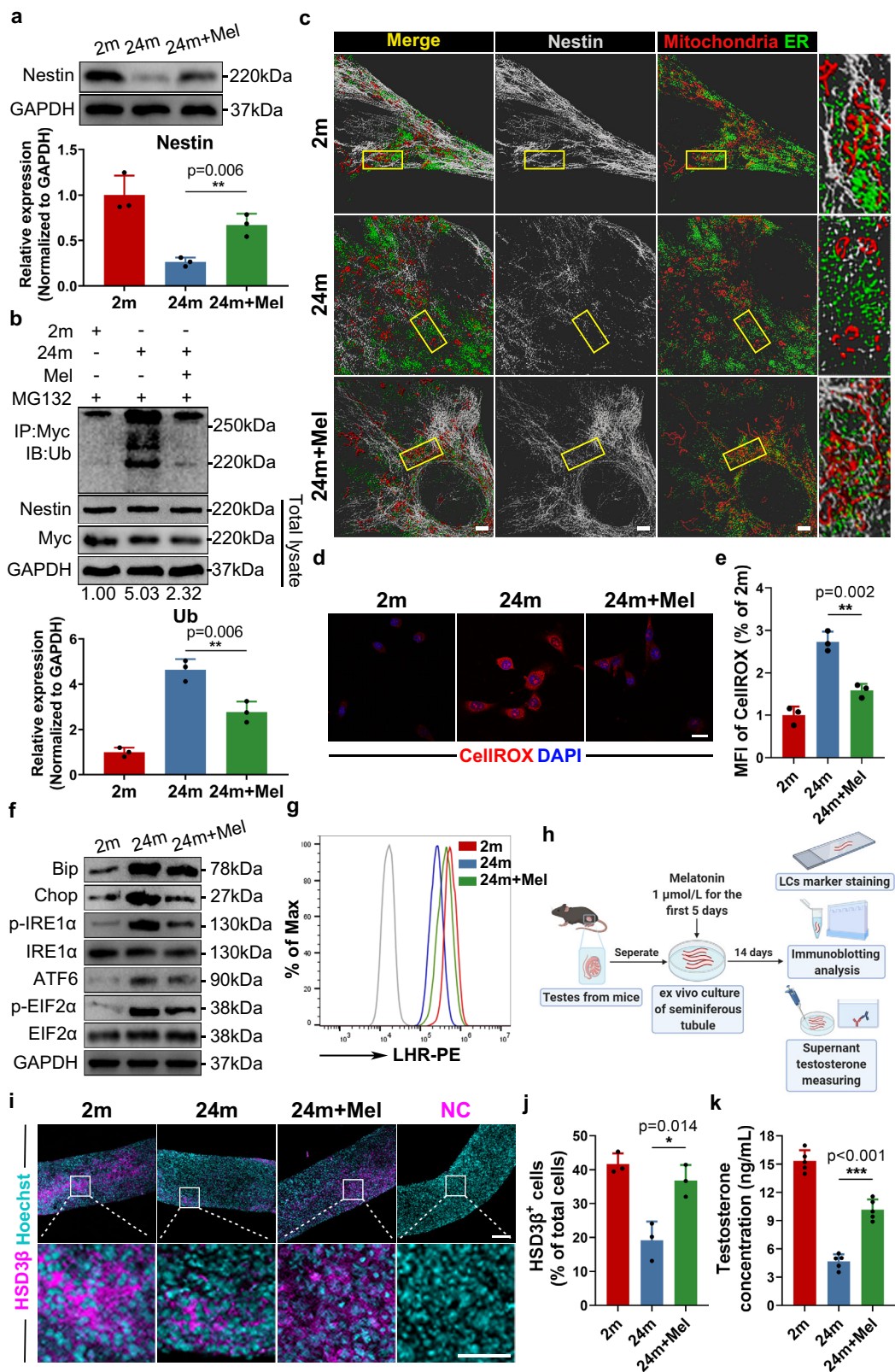

**LC differentiation of Nes-GFP+ cells in vitro**. Nes-GFP + cells were cultured in fresh differentiation-inducing medium containing phenol red-free DMEM/F12 and M199 medium (1:1; Gibco), 2% FBS, 10 ng/ml PDGFAA (PeproTech), 1 ng/ml LH (R&D Systems), 250 nM Smoothened Agonist HCl (SAG; Millipore), 1 μM Forskolin (Fsk; Sigma-Aldrich), 50 ng/ml insulin-like growth factor 1 (IGF1, PeproTech), and 5 μg/L insulin-transferrin-sodium selenite (Sigma), and they were incubated for 14 days[15].

**Culture of seminiferous tubules**. Seminiferous tubules were isolated and cut into equal lengths (5 cm)and then they were cultured in 24-well plates in phenol red-free DMEM/F12 and M199 medium (1:1; Gibco) supplemented with 0.1% BSA, LH (10 ng/mL), penicillin/streptomycin (100 U/mL and 100 μg/mL) and 5 μg/L insulin-transferrin-sodium selenite (Sigma) for up to 2 weeks at 37 °C in a humidified 5% $CO_2$ water-jacketed incubator, according to the established protocol[84].

**Fig. 5 Targeting Nestin degradation restores intracellular homeostasis and differentiation capacity in aged SLCs. a** Western Blot analysis and quantification of Nestin expression in primary SLCs from 2 months old, 24 months old, and melatonin-treated 24 months old mice. ($n = 3$ biological repeats for each group; Unpaired $t$ test). **b** Co-IP of Myc-Nestin and ubiquitin in primary SLCs after treatment with MG132. Ubiquitination level are quantified by measuring the gray-scale value of the whole lane of the Western Blot band. Relative expression of ubiquitinated Nestin (normalized to GADPH) were normalized to NTC group and are indicated at the bottom. GADPH was used as loading control. ($n = 3$ biological repeats for each group; Unpaired $t$ test). **c** Representative immunostaining pictures of Nestin, mitochondria, and ER in primary SLCs. Mitochondria and ER are marked with Mitotracker and ER tracker, respectively. Scale bar, 5 µm. **d** Representative immunostaining pictures of intracellular ROS stained with CellRox in primary SLCs. Scale bar, 20 µm. **e** Quantitative analysis of the mean fluorescence intensity of CellRox in (**e**). ($n = 3$ biological repeats for each group; Unpaired $t$ test). **f** Western Blot analysis of ER stress-related protein expression in primary SLCs. **g** Flow cytometry analysis of LHR expression in LCs induced from primary SLCs at day 9. **h** Schematic illustration of the experimental procedure for isolating seminiferous tubules from testes of mice and inducing differentiation into LCs with melatonin treatment. Created with BioRender.com. **i** Representative immunostaining pictures of seminiferous tubules after being induced to differentiation at day 14. LCs are identified as HSD3β + cells. Scale bar, 100 µm for original pictures and 50 µm for enlarged pictures. **j** Quantitative analysis of the percentage of HSD3β + cells of total cells in seminiferous tubules after induced differentiation at day 14. ($n = 3$ biological repeats for each group; Unpaired $t$ test). **k** Measurement of testosterone concentration in the medium of seminiferous tubules after induced differentiation at day 14. ($n = 3$ biological repeats for each group; Unpaired $t$ test). Two-sided comparison; All data are mean ± SD; Error bars represent SDs. *$p < 0.05$, **$p < 0.01$, ***$p < 0.001$; Uncropped western blots and source data are provided as a Source Data file. See also Supplementary Figs. 8–13.

**RNA isolation and Real-time quantitative PCR**. Total RNA was extracted from testis tissue or cells using the TRIzol reagent (Molecular Research Center, Inc.) according to the manufacturer's protocol. Quantification was performed with a NanoDrop 8000 spectrophotometer and 1 µg of total RNA was used to reverse transcription with a RevertAid First Strand cDNA Synthesis Kit (Thermo Fisher Scientific, K1622). cDNAs were used as the template for real-time quantitative PCR (qPCR) reactions with the FastStart Essential DNA Green Master Mix (Roche, 06924204001). All samples were run in triplicate and the results were normalized to 18 S rRNA as relative mRNA levels. The primers designed and used for qPCR are described in Supplementary Table 2.

**Western blot**. For Western blotting analysis, cell lysates were collected and washed twice with cold phosphate-buffered saline (PBS). Then, the cells were directly lysed in 1X RIPA buffer, and centrifuged at $10,000 \times g$ for 10 min at 4 °C to remove cell debris. Each supernatant was recovered as a total cell lysate. After total protein concentration was assessed with BCA Protein Assay Kit (Thermo), equal amount of protein was resolved by SDS polyacrylamide gel electrophoresis and then electrotransferred to a 0.45 µm pore-sized polyvinylidene difluoride membrane (Millipore, Darmstadt, Germany). The target proteins were immunoblotted with specific antibodies. The primary and secondary antibodies used can be found in the Supplementary Table 2. Chemiluminescent substrate (Millipore) was used for detecting the signaling intensity. Bands from at least three independent blots were quantified using Image J software.

**Immunoprecipitation**. Cells were lysed with a dounce homogenizer in 1 ml LP2 lysis buffer (20 mM Tris-HCl (pH 7.7), 100 mM NaCl, 10 mM NaF, 20 mM β-glycerophosphate, 5 mM MgCl₂, 0.1% Triton X-100, 5% glycerol), supplemented with benzonase (30 U/l; Santa Cruz) and complete protease inhibitor cocktail (Roche), and incubated at 4 °C for 1 h. Subsequently, lysis reactions were terminated by centrifugation at $16,000 \times g$ for 30 min, resulting in lysates. 1 ml of lysate with 10 µl of antibody-carrying beads rotated for 4–12 h at 4 °C. Precipitated products were then washed with lysis buffer and eluted with a 2.5× SDS protein gel loading solution containing 10% β-mercaptoethanol for Western blotting. Related antibodies used can be found in Supplementary Table 2.

**Flow cytometric assay**. Testicular cells derived from Nestin-Cre$^{ERT2}$; Rosa26$^{RFP}$ and Nestin-Cre$^{ERT2}$; Cas9-tdTomato mice were washed with FACS buffer (1% bovine serum albumin in PBS containing 0.5 mM EDTA) twice. Cells were stained with primary and secondary antibodies in the dark on ice at 4 °C. Cells were then washed twice with FACS buffer and run on CytoFLEX (Beckman Coulter, USA). Data were analyzed using the Flow Jo software (Tree Star Inc., Ashland, Oregon).

**Annexin V/PI flow cytometry analysis**. After indicated treatments, SLCs were incubated with 200 µM H₂O₂ for 6 h and harvested by centrifugation. Subsequently, both suspended and attached cells were collected gently and stained with annexin V/propidium iodide (PI) assay kit (BIOSCI BIOTECH, Shanghai, China) according to the manufacturer's instruction. Samples were run on a CytoFLEX (Beckman Coulter, USA) and the data were analyzed using the Flow Jo software (Tree Star Inc., Ashland, Oregon).

**Immunohistochemistry (IHC) and Immunofluorescent staining (IF)**. For IHC staining, mouse testis tissues were fixed with 4% neutral-buffered paraformaldehyde.

Paraffin-embedded tissues were sectioned by 5 mm on a Leica RM2255 rotary microtome, which was subjected to immunostaining using an UltraSensitiveTM SP (Mouse/Rabbit) IHC Kit (MXB, KIT-9710). Following deparaffinization and antigen retrieval, the testis tissue specimens were incubated overnight at 4 °C with corresponding antibodies listed in Supplementary Table 2. Signal amplification and detection were performed using a DAB system according to the manufacturer's instructions (MXB, MAX-001).

For immunofluorescent staining, the tissue sections and cells were incubated with antibodies listed in Supplementary Table 2, stained sections were imaged using a Zeiss 800 Laser Scanning Confocal Microscope, a Zeiss 880 Laser Scanning Confocal Microscope with Airyscan, Dragonfly CR-DFLY-202 2540 and a Nikon A1R N-SIM N-STORM Microscope.

**SA-β-gal staining**. According to the manufacturer's protocol of senescence-associated β-galactosidase kit (Beyotime, C0602), SLCs seeded in 12-well plates were washed twice with PBS and fixed in 4% paraformaldehyde for 15 min. After washing, SLCs were incubated overnight with the working solution of β-galactosidase plus X-Gal at 37°C. The senescent cells were observed under an optical microscope (Leika, DMi8) and counted from three random fields of vision.

**Histology analysis**. After sacrifice, mouse testis and cauda epididymidis tissues were perfused with 4% paraformaldehyde and then subjected to paraffin embedding. Paraffin-embedded tissue sections (4 µm) were cut and stained with hematoxylin and eosin (H&E). The sections were analyzed, and images were captured using a Zeiss AxioScan.Z1 microscope.

**Counting Leydig cells**. To count CYP11A1 + or HSD3β + or HSD17β3 + Leydig cells, a fractionator technique was used for the above tissue-array sections according to established protocol[85]. In brief, under the live image of a digital camera with a ×10 objective and fixed point of the "upper" sections, we counted cells of the total microscopic field. We calculated the total number of Leydig cells by multiplying Leydig cell number counted in a known fraction of the testis by the inverse of the sampling probability.

**Mitochondria and endoplasmic reticulum staining and analysis**. The mitochondrial structural network was detected by 200 nM Mitotracker Red and ER tracker Green (Cell Signaling Technology) in a culture medium at 37 °C for 30 min. Then, the cells were fixed with 4% paraformaldehyde, permeabilized in 0.2% Triton X-100 and then stained with other antibodies. Cells were imaged using a Zeiss 880 Laser Scanning Confocal Microscope with Airyscan and Nikon A1R N-SIM N-STORM Microscope. Acquired images for mitochondria morphology were analyzed[16]. The colocalization indexes Pearson's and Manders' coefficients were calculated with the ImageJ colocalization analysis plugin according to previous articles[86,87]. MERCs were also reconstructed and analyzed via surface-surface contact site area tool by Imaris (Bitplane) followed the manufacturer instructions.

**In situ proximity ligation assay**. Proximity ligation was performed in Duolink® Blocking Solution–fixed tissue or cells. According to the established protocol[88,89], samples were incubated with specific primary antibodies to the proteins to be detected. Duolink® PLA Probe were added to the reaction and incubated. Two oligonucleotides and ligase were the ligation solution. In this assay, the oligonucleotides hybridize to the two proximity ligation probes and

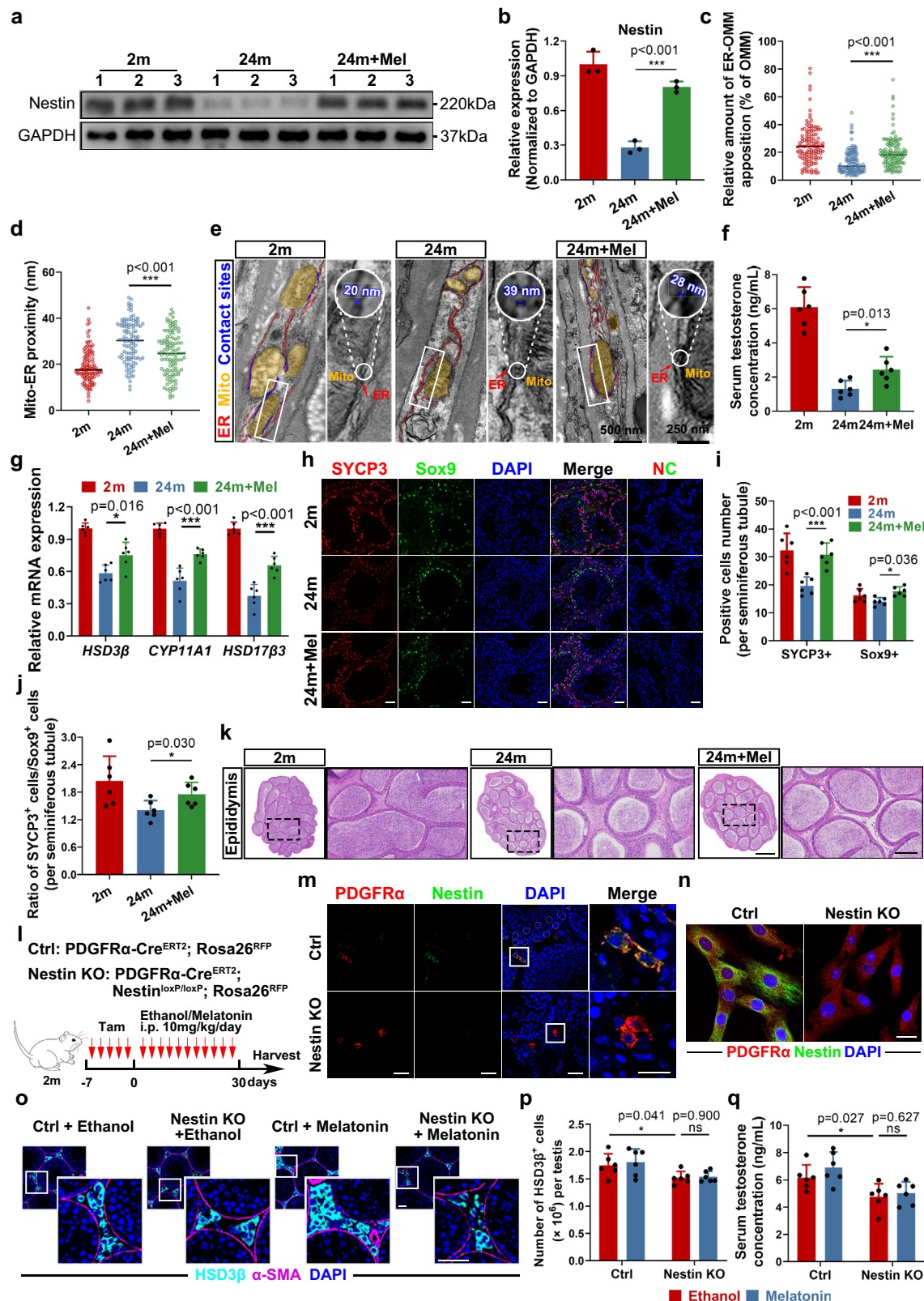

join to a closed-loop if they are in close proximity. Amplification solution, consisting of nucleotides and fluorescently labeled oligonucleotides, was added together with polymerase. The oligonucleotide arm of one of the proximity ligation probes acts as a primer for "rolling-circle amplification" using the ligated circle as a template, and this generates a concatemeric product. Fluorescently labeled oligonucleotides hybridize to the rolling circle amplification product (Duolink; SigmaAldrich). The proximity ligation signal was

shown as a distinct fluorescent spot and was analyzed by confocal microscopy. Control experiments included routine immunofluorescence staining of the proteins of interest under identical experimental conditions. PLA spots were counted in cell lines using software Blobfinder version 3.2[90]. PLA scores were calculated by normalizing the number of PLA spots counted in each sample to the average number of PLA spots counted in the control sample, which was set to 100.

**Fig. 6 Treatment with Melatonin increases testosterone level and promotes spermatogenesis in old mice. a**, **b** Western Blot analysis and quantification of Nestin expression in the testes from different age group. ($n = 3$ biological repeats for each group; Unpaired $t$ test). **c**, **d** Analysis of TEM from (**e**). Percentage of mitochondria coverage covered by ER ($n = 115$ cross-sections from 5 mice for each group) and mito-ER proximity were quantified ($n = 110$ cross-sections from 6 mice for each group); Unpaired $t$ test. **e** Representative TEM images of testicular mesenchyme. (mitochondria, yellow; ER, red; contact sites, blue). Scale bar, 500 nm for original pictures and 250 nm for enlarged pictures. **f** Measurement of serum testosterone concentration. ($n = 6$ biological repeats for each group; Unpaired $t$ test). **g** qPCR analysis of relative mRNA expression of testosterone production-related genes in the testes ($n = 6$ biological repeats for each group; Unpaired $t$ test). **h** Representative immunostaining pictures of SYCP3-positive cells and Sox9-positive Sertoli cells in seminiferous tubules of testes. Scale bar, 40 μm. **i** Quantification of the number of SYCP3-positive cells and Sox9-positive Sertoli cells per seminiferous tubule. ($n = 6$ biological repeats for each group; Multiple $t$ tests). **j** Quantification of the ratio of SYCP3-positive cells vs Sox9-positive Sertoli cells per seminiferous tubule. ($n = 6$ biological repeats for each group; Unpaired $t$ test). **k** Representative H&E staining pictures of cauda epididymis. Scale bar, 500 μm for original pictures and 50 μm for enlarged pictures. **l** Schematic of the experimental animal model for conditionally knocking out Nestin in PDGFRα + cells. Nestin KO: Nestin knockout. **m** Representative immunostaining pictures of Nestin in testicular mesenchyme of Control and Nestin KO mice. Scale bar, 40 μm for original pictures and 20 μm for enlarged pictures. **n** Representative immunostaining pictures of Nestin in primary SLCs from Control and Nestin KO mice. Scale bar, 20 μm. **o** Representative immunostaining pictures of testicular mesenchyme from Control and Nestin KO mice, treated by ethanol or melatonin. LCs are identified as HSD3β + cells. Scale bar, 40 μm for original pictures and 40 μm for enlarged pictures. **p** Quantification of LCs cell numbers in (L). ($n = 6$ biological repeats for each group; Multiple $t$ tests). **q** Measurement of serum testosterone level of Control and Nestin KO mice, treated by ethanol or melatonin. ($n = 6$ biological repeats for each group; Multiple $t$ tests). Two-sided comparison; All data are mean ± SD; Error bars represent SDs. *$p < 0.05$, ***$p < 0.001$, ns, $p > 0.05$; Uncropped western blots and source data are provided as a Source Data file. See also Supplementary Fig. 14.

**Transmission electron microscopy**. Cells grown in chamber slides were fixed in 2.5% glutaraldehyde in PBS for 2 h at room temperature, then stored at 4 °C overnight before processing. Thin sections on grids were observed in a Tecnai 12 BioTwin transmission electron microscope (FEI) at 120 keV. Images were acquired with a charge coupled device camera (AMT) according to previous article[22].

For tissue microscopy, the testis was isolated from mice, cut in 1 mm thick sagittal sections and small portions of upper layers of the cortex were dissected for further processing (for injured cortices, the examined area was dissected according to the location of the lesion track). For EPON embedding, the fixed tissue was washed with 0.1 M sodium cacodylate buffer, incubated with 2% $OsO_4$ in 0.1 M cacodylate buffer (Osmium, Science Services; Caco Applichem) for 2 h at 4 °C and washed again three times with 0.1 M cacodylate buffer. Subsequently, tissue was dehydrated using an ascending ethanol series with 15 min incubation at 4 °C in each EtOH solution. Tissues were transferred to propylene oxide and incubated in EPON (Sigma-Aldrich) overnight at 4 °C. Tissues were placed in fresh EPON at RT for 2 h, followed by embedding for 72 h at 62 °C. Ultrathin sections of 70 nm were cut using an ultramicrotome (Leica Microsystems, UC6) with a diamond knife (Diatome, Biel, Switzerland) and stained with 1.5% uranyl acetate at 37 °C for 15 min and lead citrate solution for 4 min. Electron micrographs were taken with a JEM-2100 Plus Transmission Electron Microscope (JEOL), equipped with Camera OneView 4 K 16 bit (Gatan) and software DigitalMicrograph (Gatan).

For analysis, the length of each mitochondrion was determined in ImageJ following a manual drawing of single organelles. The extent of mitochondria-ER contact sites (defined as sites of contact within a reciprocal distance of 100, 50, and 25 nm) is calculated as the ratio of ER-OMM apposition length/total OMM length. Minimal mitochondria-ER proximity is determined by measuring the minimal distance between mitochondrion and ER within a mitochondria-ER contact site. All parameters obtained from one field of view (usually containing several mitochondria and multiple contact sites) were averaged together.

**ROS Generation**. Intracellular ROS levels were determined by DHE staining according to the manufacturer's instructions. Briefly, treated cells were incubated in DMEM containing 10 μmol/L DHE (Sigma-Aldrich) at 37 °C for 30 min at dark. After washing, labeled cells were evaluated using flow cytometry. Data were analyzed by Flow Jo Software (Tree Star Inc., Ashland, Oregon). 150 μM antimycin treatment was used as the positive control.

CellROX™ Deep Red Reagent (Thermo Fisher Scientific) was also used to stain Intracellular ROS levels according to manufacturer's instructions. Briefly, CellROX Reagent was added at a final concentration of 5 μmol/L and incubated for 30 min at 37 °C. After fixing with 3.7% formaldehyde, cells were analyzed by confocal microscope. 200 μM TBHP treatment was used as the positive control.

The levels of mitochondrial ROS were incubated in SLC medium using the fluorescent probes MitoSOX™ Red (Invitrogen) for 30 min. After washing, labeled cells were evaluated using flow cytometry. Data were analyzed by Flow Jo Software. (Tree Star Inc., Ashland, Oregon).

**Measurement of total GSH and GSSG**. Total glutathione (GSH + GSSG, tGSH) and GSSG levels were determined by the method of Rahman et al[91]. Briefly, cells are harvested and washed twicewith PBS. Next, we centrifuged and discarded the

supernatant, resuspended the cells in 1 ml of ice-cold extraction buffer, containing 0.1% Triton-X and 0.6% sulfosalicylic acid in KPE (0.1 M potassium phosphate buffer with 5 mM EDTA disodium salt, pH 7.5). After that, sonicate the suspension in icy water for 2–3 min with vortexing every 30 seconds to obtain cytosolic fractions. The rates of 5′-thio-2-nitrobenzoic acid (TNB) formation were calculated, and the tGSH and GSSG concentrations in the samples were determined by using linear regression to calculate the values obtained from the standard curve. The GSH concentration was determined by subtracting the GSSG concentration from the tGSH concentration. The values were normalized by protein concentration. All samples were run in duplicate. All reagents used in this assay were purchased from Sigma-Aldrich.

**Thioredoxin reductase activity**. The activity of thioredoxin reductase was measured using the Thioredoxin Reductase Assay kit (Sigma-Aldrich) according to the manufacturer's instructions. In brief, cytosolic fractions were added to 96 well plate and then mixed with working buffer, 100 mm DTNB. Either 1× assay buffer or thioredoxin reductase inhibitor was added to the wells. Enzymatic activity was determined by subtracting the time-dependent increase from total activity in absorbance at 412 nm every 10 s for 2 min in a spectrometer in the presence of the thioredoxin reductase inhibitor. All samples were run in duplicate.

**Mitochondrial membrane potential assessment**. For measurement of mitochondrial membrane potential, cells were stained with 200 nM Mitotracker Green (Thermo Fisher Scientific), 200 nM of tetramethylrhodamine, ethyl ester (TMRE, Thermo Fisher Scientific) at 37 °C for 30 min. Mitochondrial depolarization was calculated as % of the area of green mitochondria compared to that of all the mitochondria visualized on Mitotracker Green and TMRE merged images. Data are presented as depolarized mitochondria (green) / total mitochondria (green + yellow) × 100%. 20 μM FCCP was used as the positive control.

**Metabolism assays**. Intracellular ATP production was measured using a CellTiter-Glo Luminescent Cell Viability Assay kit. (Promega, Madison, WI, USA), per the manufacturers' instructions. Extracellular lactate production and intracellular glucose uptake were measured using lactate and glucose assay kits (Promega, Madison, WI, USA), respectively.

**Testosterone concentration assay**. The cell culture supernatants and sera were collected at each experimental time point for the quantitative determination of testosterone. Testosterone levels were measured using a commercially available ELISA kit (R&D Systems, Minneapolis, MN, USA), according to the manufacturer's instructions.

**Semen collection and analysis**. The epididymides were collected and used for semen analysis according to a previous article[92]. In brief, each fresh epididymis was cut into small pieces and dispersed in 0.5 ml of F12 medium (GIBCO) containing 0.1% bovine serum albumin (Invitrogen) prewarmed to 37 °C and incubated for 15 min to facilitate the transmigration of sperm from the epididymis. Each epididymal sperm suspension was subjected to sperm counting and sperm motility analyses by computer-aided semen analysis (CASA) system. In brief, cauda epididymides were removed and incised with micro scissors, and incubated in 0.5 mL buffer containing 0.5% BSA (Sigma) for 15 min at 37 °C to allow for sperm release.

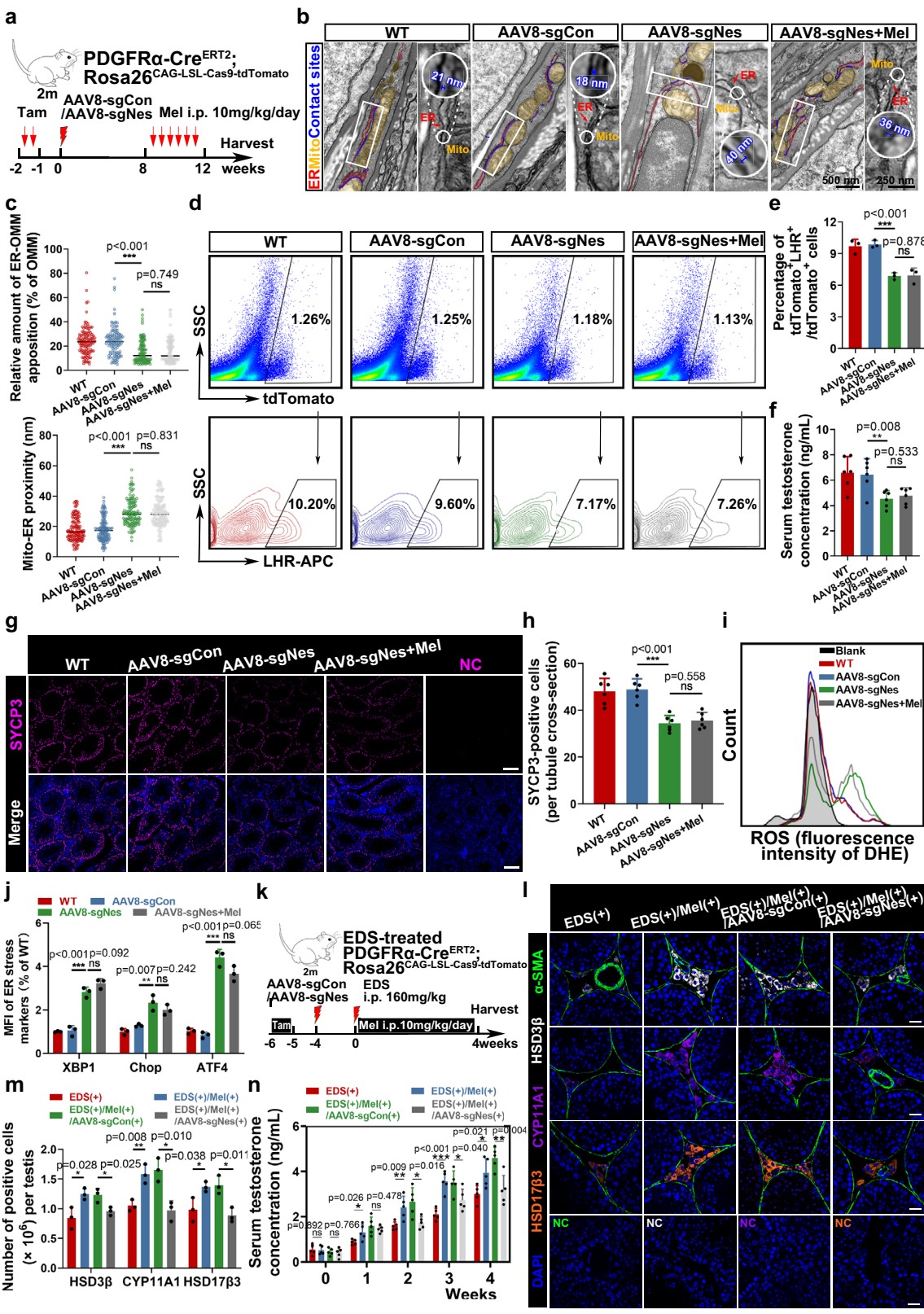

The tissue was removed, and sperm samples were diluted and analyzed using a Hamilton Thorne's Ceros II system. At least six fields were assessed for each sample, and the sperm concentration and percentages of motile and progressively motile spermatozoa were determined.

**Statistics and reproducibility.** All experiments were carried out with at least three biological replicates and showed successful reproducibility. All data are reported as

the mean ± SD of at least three independent experiments. Sample sizes are all presented in the figure legends. Statistical analysis between two groups was performed using unpaired *t*-test. Statistical analysis between multiple groups was performed by one-way ANOVA, with Tukey's multiple comparison test. All data were analyzed using Prism software (GraphPad Software). A two-sided p-value < 0.05 was considered to be statistically significant. The level of significance defined as $p < 0.05$ (*), $p < 0.01$ (**), $p < 0.001$ (***).

**Fig. 7 AAV-mediated downregulation of Nestin separates MERCs and diminishes Melatonin's effect in attenuating male reproductive ageing.**
**a** Schematic of the experimental animal model for conditionally knocking out Nestin in testis. **b** Representative TEM images of testicular mesenchyme. (mitochondria, yellow; ER, red; contact sites, blue). Scale bar, 500 nm for original pictures and 250 nm for enlarged pictures. **c** Analysis of TEM from (**b**). Percentage of mitochondria coverage covered by ER ($n = 115$ cross-sections from 4 mice for each group) and mito-ER proximity were quantified ($n = 130$ cross-sections from 5 mice for each group; Unpaired $t$ test). **d** Flow cytometry of the percentage of tdTomato+LHR+ cells in vivo. **e** Quantification of percentage of tdTomato+LHR+ cells in total tdTomato+ cells in (d) ($n = 3$ biological repeats for each group; Unpaired $t$ test). **f** Serum testosterone level of wild type, AAV8-sgCon, AAV8-sgNes, and AAV8-sgNes+Mel groups. ($n = 6$ biological repeats for each group; Unpaired $t$ test). **g** Representative immunostaining pictures of SYCP3-positive cells in seminiferous tubules of testis. Scale bar, 100 μm. **h** Quantification of SYCP3-positive cells per tubule cross-section. ($n = 6$ biological repeats for each group; Unpaired $t$ test). **i** Flow cytometry of intracellular ROS level stained with DHE of tdTomato+ cells in vivo. **j** Quantification of mean fluorescence intensity of ER stress markers in Supplementary Fig. 15j. ($n = 3$ biological repeats for each group; Multiple $t$ tests). **k** Schematic of the chemical-damaged animal model treated with EDS and conditionally knocking out Nestin in testis. **l** Representative immunostaining pictures of testicular mesenchyme from EDS( + );EDS( )/Mel(+);EDS( )/Mel(+)/AAV8-sgCon(+);EDS( )/Mel(+)/AAV8-sgNes(+) groups 4 weeks after EDS treatment. LCs are identified as HSD3β + /CYP11A1 + /HSD17β3 + cells. Scale bar, 20 μm. **m** Quantification of LCs cell numbers in (**l**) with different markers. ($n = 3$ biological repeats for each group; Multiple $t$ tests). **n** Measurement of serum testosterone level. ($n = 5$ biological repeats for each group; Multiple $t$ tests). Two-sided comparison; All data are mean ± SD; Error bars represent SDs. $*p < 0.05$, $**p < 0.01$, $***p < 0.001$, ns, $p > 0.05$; Source data are provided as a Source Data file. See also Supplementary Fig. 15.

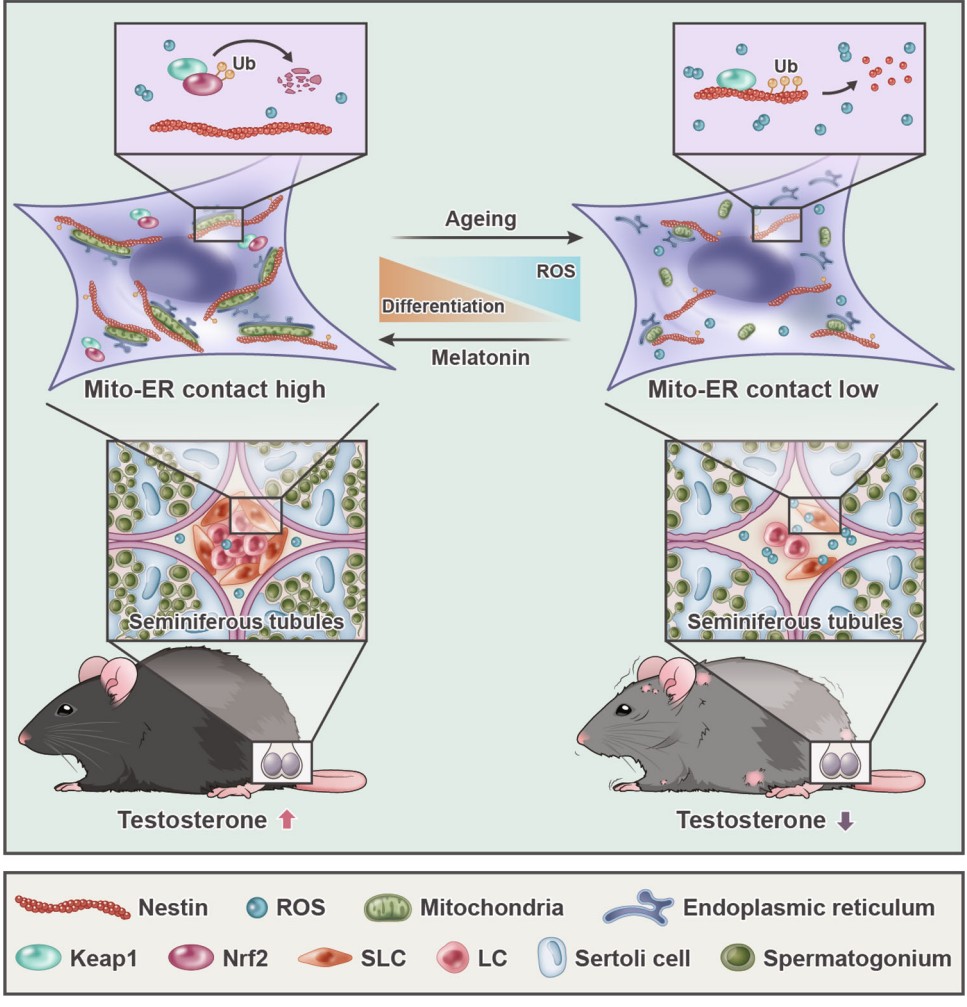

**Fig. 8 Schematic illustration of Nestin-mediated MERCs loss during reproductive ageing and melatonin's anti-senescence effect.** Loss of intermediate filament Nestin compromises the differentiation capacity of SLCs via separating MERCs during senescence. Pharmacological intervention by melatonin restores Nestin-dependent MERCs, improving SLC differentiation capacity and alleviating male reproductive system ageing.

**Reporting summary**. Further information on research design is available in the Nature Research Reporting Summary linked to this article.

corresponding author (S.A.S.) upon request. All uncropped western blots and data values for all figures are provided in the source data file. Source data are provided with this paper.

## Data availability
The authors declare that data supporting the findings of this study are available within the article and its Supplementary information files or from the

## Code availability
All software and bioinformatic tools used in the present study are publicly available.

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

## Acknowledgements

The authors would like to thank the Core Facility of Center, Zhongshan School of Medicine, Sun Yat-Sen University for providing instruments for a Zeiss 800 Laser Scanning Confocal Microscope, a Zeiss 880 Laser Scanning Confocal Microscope with Airyscan, and a Nikon A1R N-SIM Microscope. The National Key Research and Development Program of China, Stem cell and Translational Research (2021YFA1100601, 2018YFA0107200, 2017YFA0103403, and 2017YFA0103802), the National Natural Science Foundation of China (81730005, 31771616, 81802402, 81971372, 32130046, 32170799, 82101367, and 82170613), the Key Research and Development Program of Guangdong Province (2017B020231001, 2019B020234001, 2019B020236002, and 2019B020235002), the Natural Science Foundation of Guangdong Province (2022A1515012370), Key Scientific and Technological Program of Guangzhou City (201803040011), and the Research Start-up Fund of the Seventh Affiliated Hospital, Sun Yat-sen University (393011).

## Author contributions

Conceptualization, J.W., S.Y., X.W., and W.D.; Methodology, J.W., S.Y., X.W., W.D., B.W., and Y.W.; Investigation, S.Y., X.W., W.D., B.W., and Y.Q.; Formal Analysis, S.Y., X.W., Y.G., J.C., and Y.H.; Writing, S.Y., X.W., W.D., and X.L., Funding Acquisition, J.W.; Supervision, J.W. and S.Y.

## Competing interests

The authors declare no competing interests.
