## [Peer Review File · Nature Communications]

Nestin-dependent mitochondria-ER contacts define stem Leydig cell differentiation to attenuate male reproductive ageingReviewers' comments:

Reviewer #1 (Remarks to the Author):

In their manuscript entitled "Nestin-dependent mitochondria-ER contact defines stem Leydig cell differentiation to attenuate male reproductive aging", Senyu Yao and colleagues reported that differentiation capacity of stem Leydig cells (SLCs) is decreased along with aging, and that it results in deficiency of testosterone production. They also showed that melatonin reverses SLC differentiation capacity by restoring Nestin-dependent mitochondria-ER contact.

It has been known that mito-ER contact play important role in steroidogenesis, and loss of mito-ER contact occurs in various kinds of stem cells after senescence. In their precious work, they reported that Nestin influences mitochondrial function, and regulate the maintenance and stemness in neural stem/progenitor cells. In the current study, they found that in aged SLCs, loss of intermediate filament Nestin separates mito-ER contact, and showed that it is the cause of decreased differentiation capacity of aged SLCs. They also identified that Nestin regulates mito-ER contact by competing with Nrf2 in binding Keap1. Furthermore, they found that melatonin, which has been known to protect mitochondria and influence Nrf2 and Keap1, can restore mito-ER contact and testosterone production. This study is novel in showing that mito-ER contact is involved in aging of SLCs, and most intriguingly, that melatonin can reverse the phenotype of aged SLCs. Overall the manuscript is well written and publication is recommended after modification.

Specific comments:

- (1) Overall, the letters in Figures are too small.
- (2) P11 line 11; It should be stated that melatonin is treated in vivo.
- (3) P13 line 14 "that more spermatogonia stained with SYCP3 were"; SYCP3 is also expressed in spermatocyte and they cannot be distinguished without staining by spermatogonia-specific marker. Change expression or use spermatogonia-specific marker.
- (4) P17 line 2 "ER-mito contact"; Although the term "Mito-ER" is used throughout manuscript, is there any difference between these two terms?
- (5) P30 line 3; There is no explanation about what CASA system is. Clarify source.
- (6) Figure 3H; Label should be corrected to small character.
- (7) Figure 4i, j: Need more description in legends. Also quantification of the data is preferable. Legends of Figure 4i does not match the figure.
- (8) Figure 5c: Quantification of the data is preferable.
- (9) Figure 5e: Counting number of SYCP3 per tubules in section is not fair, because tubule size varies between samples and angle of section. I recommend showing the ratio of SYCP3+ cells vs. Sertoli cells.
- (10) Figure 6n: please show the age of Nestin KO mice.
- (11) Statistics; Is there any reason for showing data by SD instead of SEM throughout the study?
- (12) Figure 8; I guess that graphical abstract will be better if left and right are reversed, showing that young mice in the left changes into the status of aged mice in the right side, and melatonin can reverse the phenotype. Also "ER-Mito" is used in the figure, but in legend it is "Mito-ER". Please use the same term throughout the manuscript.

Reviewer #2 (Remarks to the Author):

Yao et al. propose that ER-mitochondria contacts are a key structure that promotes the survival and functionality of Leydig cells, responsible for the maintenance of the male reproductive system. During

aging, they observe decreased viability that coincides with decreased expression of Nestin and a decreased formation of ER-mitochondria contacts. The study currently fails to provide good assays for the claimed changes of ER-mitochondria contacts. The authors also provide experiments for pathways and proteins that are poorly rationalized, ignoring other more important pathways known to play a role for ER-mitochondria contacts. Specifically, they pick Nestin and claim it is critical for ER-mitochondria contacts but fail to provide evidence for this hypothesis. Similarly, the evidence for the association of melatonin in the rescue of this structure and the overall phenotype is coincidental at best. Therefore, the manuscript contains a lot of conjecture and correlative data but fails to provide mechanistic insight.

Specific points

1. The manuscript is riddled with spelling and grammar mistakes, which are too many to list. The authors must consult with a native English speaker to edit the text. In particular, they must pay close attention to the proper usage of articles and singular/plural.
2. The characterization of ER-mitochondria contacts is not properly done. The authors show PLA for Grp75 and CRT. However, these two proteins do not co-localize and can therefore not be used for this purpose, the latter being restricted to the ER lumen, the former found on the surface of mitochondria.
3. In the past, actin had been identified as a cytoskeletal factor critical for ER-mitochondria contacts and their role for mitochondrial morphology. The authors should put their study into context with Korobova et al. In particular, what happens to INF2 in their system? How are actin branching factors affected?
4. The measurement of ROS in Figure 3a is confusing and I do not detect significant changes from the flow cytometry plots. What was measured? The number of positive cells? How were they chosen? Or their signal? How were ROS-positive cells identified? This must be indicated on the plot.
5. The choice of Nestin as a cytoskeletal component undergoing changes is arbitrary and lacks a good rationale and controls. What happens to other intermediate filaments? The role of Nestin for ER-mitochondria contacts is not demonstrated with the data presented. Also, the authors do not investigate the functional significance of Nestin for ER-mitochondria contacts. What is its function for this structure? This remains a mystery. Later in the manuscript, Nestin amounts are correlated with Leydig cell presence and proliferation. It remains unclear whether this is one of many factors that could lead to problems during aging and even more, whether it is a key factor.
6. Similarly, melatonin is one of very many antioxidants that could undergo changes during aging. Why was this chosen over others? What happens to proteins of the thioredoxin family, known regulators of ER-mitochondria contacts (e.g., ERdj5, TMX1)? This is a critical flaw, since the authors are not able to rescue ER stress, as shown by the persistent elevated levels of BiP, suggesting other mechanisms are at play.
7. There are no functional links between melatonin and ER-mitochondria contacts in the manuscript other than a non-quantified fluorescence microscopy in Figure 5d. Of course, a clear association of melatonin availability with ER-mitochondria contact formation would require massive amounts of work.

Reviewer #3 (Remarks to the Author):

The overall objective of this study is to analyse the relationship between SLC differentiation and Male

reproductive system aging. It is known that SLC differentiation relies on lipids exchange among different subcellular compartments, and that lipids homeostasis is maintained by mitochondria-ER crosstalk. In this work, the authors hypothesize that loss of nestin-mediated Mitochondria-ER contact sites underlies the progressive decline of SLC differentiation associated with aging, and show that the phenotype can be reverted upon melatonin treatment.

This work has the potential to highlight a mechanism involved not only of SLC differentiation, but, in general, of stem cells: the demonstration that contact sites among organelles are key drivers in the differentiation is likely to be expanded to other cell models/tissues in the years to come, due to their fundamental role in the maintenance of cell physiology.

However I have several concerns (listed hereafter) regarding the experimental procedures that have been described, and that need to be addressed to make the paper suitable for publication:

- 1) Mitochondrial Morphology: could the author provide the average value which was measured in nanometers, and not as % of 2 μ m (or, could they add the actual values at least in the supplementary materials)?
- 2) What is the average mito-ER contact sites width in the different conditions (nanometers)? What about the length? These parameters can be determined by TEM images analysis.
- 3) It is surprising to me that PLA works with a protein residing in the lumen of ER shows a positive signal (normally for contact sites, proteins residing on the surface of the 2 organelles are used); is there any publication in which calreticulin has been described at MAMs? PLA is used for detection of protein-protein interactions at distances <40 nm. Could the author address this point?
- 4) I have several concerns related to the TMRM analysis. First, was the dye used in quenching mode or not? 200nM seems a high concentration. According to the methods section, no Cyclosporin H was used. Controls (as FCCP, or oligomycin, or both) are missing.
- 5) DHE measurement: a positive control (for example, antimycin) is missing
- 6) Figure 3 is entitled: "Less mito-ER contact results in mitochondrial dysfunction and ER stress in aged SLCs". However, the authors do not provide a formal demonstration of this point. At variance, figure 3 provide evidence that conditions associated with lower mitoER contact sites (that is, 12 and 24 months) are also characterised by changes in the expression levels of several ER stress-related genes and anti-oxidative genes. To really demonstrate that less contact sites results in mito dysfunction in SLCs they should: (i) characterise which MAMs resident protein is downregulated during aging; (ii) change the expression of MAMs resident proteins and check the effects on ER stress and REDOX genes.
- 7) Figure 4b. Immunofluorescence is a semi-quantitative technique; the authors should confirm cytoskeletal proteins expression by Western Blot
- 8) Figure 4f: the term "nestin organization" needs to be defined better. What did the authors quantify exactly?
- 9) With reference to figure 6: what happens in this model (e.g. 24 months+mel, with respect to 24 months only) to contact sites, after treatment with Melatonin?
- 10) The parameters that authors take into account to study mito ER contacts should be consistent (they use percentages in figure 2, and proximity in nm in figure 7). Which contact sites range do the author consider in the different models proposed (I refer to Figure 2 and 7): do they always analyse contact sites <100nm? If not, why?
- 11) Extended data figure 2: control are missing (example, FCCP; additional western blots to study autophagy related protein expression, such as for example LC3 I and II; etc.)
- 12) Extended figure 5: what happens in case of treatment with melatonin of 2m old SLCs (e.g. without H2O2)?
- 13) Panel o of Extended data figure 5 needs to be incorporated in extended data figure 6.

Minor:

Not sure References are listed correctly through the text. Please check.

Figure 2: what is the Total apposition length? Is this the mean surface covered by contacts on top of mito perimeter? Please specify better.

Point-by-point response to the reviewers

Reviewer #1:

Point 1: Overall, the letters in Figures are too small.

Response: Thank you for the helpful suggestions. As suggested, we revised all the text size in **Fig.1 to Fig.7 and Extended Data Fig. 1 to Extended Data Fig. 16** in the revised manuscript for a better recognition. The font size of text with lower case will be enlarged to 8–12 points, while the font size of text with upper case (labeling different part of the figure) will be enlarged to 12-14 points.

We sincerely thank for your comments and hope the figures in our article will show clear presentation.

Point 2: P11 line 11; It should be stated that melatonin is treated in vivo.

Response: We apologize for our unclear description of the melatonin treatment. As suggested, we added “in vivo” after “melatonin treatment” in order to clarify all the melatonin treatments that appears in this paragraph are treated in vivo.

Point 3: P13 line 14 “that more spermatogonia stained with SYCP3 were”; SYCP3 is also expressed in spermatocyte and they cannot be distinguished without staining by spermatogonia-specific marker. Change expression or use spermatogonia-specific marker.

Response: We apologize for the unclear description of the cells stained by SYCP3 due to our oversight during the preparation of the original manuscript. As suggested, we revised the expression into “that the expression of meiotic marker, SYCP3 was increased in the seminiferous tubules”.

Point 4: P17 line 2 “ER-mito contact”; Although the term “Mito-ER” is used throughout manuscript, is there any difference between these two terms?

Response: We feel sorry for this inconsistency. Actually, there is no any difference

between these two terms. As suggested, we checked and corrected all the expression into “mito-ER contact” in describing the structural and functional contact between mitochondria and the endoplasmic reticulum throughout the manuscript.

Point 5: P30 line 3; There is no explanation about what CASA system is. Clarify source.

Response: Thank you for your kind advice. As suggested, we provided detailed description for CASA system in the **Methods** section, which was according to our previous article (*Theranostics*. 2020 Jul;10(19):8705-8720). The added description is attached below.

“In brief, one cauda epididymis was removed from each mouse, incised with micro scissors, and incubated in 0.5 mL buffer containing 0.5% BSA (Sigma) for 15 min at 37°C to allow for sperm release. The tissue was removed, and sperm samples were diluted and analyzed using a Hamilton Thorne’s Ceros II system. At least six fields were assessed for each sample, and the sperm concentration and percentages of motile and progressively motile spermatozoa were determined.”

Point 6: Figure 3H; Label should be corrected to small character.

Response: As suggested, the label was corrected to smaller character to 11 points in **Fig. 3h**.

Point 7: Figure 4i, j; Need more description in legends. Also quantification of the data is preferable. Legends of Figure 4i does not match the figure.

Response: We apologize for our unclear description for the co-immunoprecipitation assay in **Fig. 4i and j**. As suggested, we added more detailed descriptions to the figure legend section to make the corresponding figures understood better. Meanwhile, we provided the quantification of the data as suggested (*Proc Natl Acad Sci U S A*. 2017 Apr;114(17):4489-4494; *J Cell Biol*. 2017 May;216(5):1301-1320). These results further confirmed that the age-related loss of Nestin resulted from increased degradation (**Fig. 4i**) and Keap1 is essential in mediating the ubiquitin-proteasomal

degradation of Nestin (**Fig. 4j**). **The corresponding description and data were added as Fig. 4i and j.**

Point 8: Figure 5c: Quantification of the data is preferable.

Response: Thank you for your kind advice. Accordingly, we also provided the quantification of the data as suggested (*Proc Natl Acad Sci U S A.* **2017 Apr;114(17):4489-4494**; *J Cell Biol.* **2017 May;216(5):1301-1320**). The result showed that melatonin could successfully inhibit ubiquitin-proteasomal degradation of Nestin. **The corresponding data was added as Fig. 5c.**

Taken together, we believe that these quantifications can better explain our conclusions.

Point 9: Figure 5e: Counting number of SYCP3 per tubules in section is not fair, because tubule size varies between samples and angle of section. I recommend showing the ratio of SYCP3+ cells vs. Sertoli cells.

Response: This is really a good suggestion. Actually, we counted SYCP3-positive cells (per tubule cross-section) and percentages of SYCP3-positive tubules per section according to the **Methods** section in our previous articles (*Cell Res.* **2014 Dec;24(12):1466-85**; *Stem Cells.* **2017 May;35(5):1222-1232**). Meanwhile, as suggested, we performed additional immunofluorescent staining of SYCP3+ cells co-stained with Sertoli cells, marked by SRY-box transcription factor 9 (Sox9), and found that SYCP3+ and Sox9+ cells were decreased in the seminiferous tubules as well as their ratio during ageing, which was consistent with the results found in human (*Int Urol Nephrol.* **2014 May;46(5):879-85**), and the melatonin treatment could partly reverse the number of SYCP3+ and Sox9+ cells as well as their ratio in aged mice. These results further verified the efficacy of melatonin for spermatogenesis in vivo. **The corresponding data was added as Fig. 6h-j.**

Point 10: Figure 6n: please show the age of Nestin KO mice.

Response: Thank you for your advice. As suggested, we clarified the age of all the mouse models mentioned in the legends of **Fig. 6 and Fig. 7**. The schematic of the experimental animal models was revised in the original manuscript to make the figure clearer. **The corresponding data was presented as Fig. 6 and Fig. 7.**

Point 11: Statistics; Is there any reason for showing data by SD instead of SEM throughout the study?

Response: We thank the reviewer for bringing this point to our attention. Actually, the standard deviation (SD) measures the amount of variability, or dispersion, from the individual data values to the mean. In contrast, the standard error of the mean (SEM) measures how far the sample mean of the data is likely to be from the true population mean. Here, by providing the bar in our statistical analysis, we tend to demonstrate the error caused by imprecision in our experiments. In this regard, SD provides more reliable evidence than SEM.

Taken together, we believe that it is more appropriate and convincing to use SD rather than SEM in our statistical analysis.

Point 12: Figure 8; I guess that graphical abstract will be better if left and right are reversed, showing that young mice in the left changes into the status of aged mice in the right side, and melatonin can reverse the phenotype. Also "ER-Mito" is used in the figure, but in legend it is "Mito-ER". Please use the same term throughout the manuscript.

Response: Thank you for your precious advice. This is really a good idea. For the first concern, as suggested, we revised the graphical abstract to show more clearly that loss of intermediate filament Nestin reduces the differentiation capacity of SLCs via separating mito-ER contact during ageing and pharmacological intervention by melatonin can reverse the phenotypes. **The corresponding data was presented as Fig. 8.**

For the second concern, according to the reviewer's suggestion, we corrected all

the expression into “mito-ER contact” in describing the structural and functional contact between mitochondria and the endoplasmic reticulum throughout the manuscript.

Reviewer #2:

Point 1: The manuscript is riddled with spelling and grammar mistakes, which are too many to list. The authors must consult with a native English speaker to edit the text. In particular, they must pay close attention to the proper usage of articles and singular/plural.

Response: Thank you for pointing out our mistakes in spelling and grammar. Accordingly, the revised manuscript was proofread by native English professionals from American Journal Experts (Certificate Verification Key: 257C-2AEA-D403-3FEA-1644). We have laid emphasis on correcting articles and singular/plural forms in the revised manuscript and we believe that the revised manuscript would be much more readable.

Point 2: The characterization of ER-mitochondria contacts is not properly done. The authors show PLA for Grp75 and CRT. However, these two proteins do not co-localize and can therefore not be used for this purpose, the latter being restricted to the ER lumen, the former found on the surface of mitochondria.

Response: Thank you for your inspiring query. Actually, although calreticulin was long considered to be a Ca²⁺ binding protein that locates inside the ER lumen, it was proved that the distribution of calreticulin was also enriched in mitochondrial-associated ER membrane (MAM) (*Cell*. 2007 Nov;131(3):596-610). Moreover, the activity and action of calreticulin are regulated by other chaperones or proteins most likely occurring at the MAM of the ER (*Trends Cell Biol*. 2009 Feb;19(2):81-8), which strengthens the idea that calreticulin can be found in MAM, a membranous ER structure that is in proximity with mitochondria. These works may explain why calreticulin, a protein that resides in the lumen of ER, can be detected to be spatially close to mitochondrial membrane-bound protein.

Furthermore, it would be a better idea to use a classical protein complex, which makes ER bridge to mitochondria. As suggested, we used inositol 1,4,5-triphosphate

receptors (IP3R1) as a marker for ER and Glucose-regulated protein 75 (GRP75) as a mitochondrial marker to represent the proximity of the two organelles according to the previous protocols. The results showed that SLCs establish less mitochondria-endoplasmic reticulum contact during ageing. **The corresponding data was added as Fig. 2h-j.**

We sincerely appreciate your comments and hope the additional evidence will make our work more convincing.

Point 3: In the past, actin had been identified as a cytoskeletal factor critical for ER-mitochondria contacts and their role for mitochondrial morphology. The authors should put their study into context with Korobova et al. In particular, what happens to INF2 in their system? How are actin branching factors affected?

Response: We appreciated your inspiring comments. Actin has been identified as a classical regulatory factor of the mito-ER contact, either the change of actin polymerization/depolymerization or branching can result in the alteration of the contact between mito-ER as well as the structure and function of both organelles. Korobova et al. did a good job and found that actin polymerization through ER-localized inverted formin 2 (INF2) was required for efficient mitochondrial fission in mammalian cells (*Science*. 2013 Jan;339(6118):464-7). Actually, in our study, the staining of actin by phalloidin and western blot assay have already revealed the stability of actin between young and old SLCs (**Fig. 4a-c, 4e and 4g-h**), while there existed significant changes in the expression of Nestin. (**Fig. 4a-c, 4f and 4g-h**). Also, according to your kind advice, we performed additional experiments to detect the change of INF2 and actin branching factors as suggested (*J Cell Sci*. 2009 May;122(Pt 9):1430-40; *Mol Biol Cell*. 2008 Apr;19(4):1561-74). The results showed that there existed no significant differences in the protein expression of INF2 and actin branching factors such as Arp2/3 between young and old SLCs. These data probably indicate that nestin, rather than actin, may be the most critical cytoskeletal factor for contributions to mito-ER contact, natural SLC differentiation and senescence. Altogether, we hope the additional evidence will

make our work more convincing. **The corresponding data was added as Extended Data Fig. 5.**

Furthermore, compared with the work by Korobova et al. (*Science*. 2013 Jan;339(6118):464-7), we speculated that the contradiction might result from the **different cell types** (a human osteosarcoma cell line and a mouse fibroblast line vs. primary stem Leydig cells), **different conditions** (artificial intervention, e.g. constitutively active INF2-CAAX vs. natural state of SLC differentiation or senescence) and so on. **The corresponding information was added in the Discussion Section of the revised manuscript.**

Point 4: The measurement of ROS in Figure 3a is confusing and I do not detect significant changes from the flow cytometry plots. What was measured? The number of positive cells? How were they chosen? Or their signal? How were ROS-positive cells identified? This must be indicated on the plot.

Response: We appreciate your professional comments and we feel sorry for the unclear description for ROS generation analysis. Actually, we conducted flow cytometry in order to gate and sort out Nestin-GFP⁺ cells, which were considered as SLCs in the testis (*Cell Res.* 2014 Dec;24(12):1466-85). Next, for further analysis of intracellular ROS levels in SLCs from different age groups, we chose DHE (Dihydroethidium) as the fluorescent intracellular ROS indicator to identify ROS low or ROS high cells. (*Cell*. 2020;181(6):1307-1328.e15; *Circulation*. 2005;111(18):2347-55; *Cell*. 2015;163(3):560-9). Finally, we measured their relative mean fluorescence intensity of the signal (% of 2m) and found that there was an increase of intracellular ROS levels in SLCs with age, suggesting the disruption of mito-ER contacts during ageing could lead to impaired intracellular homeostasis. Regarding your confusion that you didn't detect significant changes, we thought that was probably because the x-axis of the flow cytometry plots was index change, where small changes could make a big difference. The actual mean fluorescence intensity of DHE of Nestin-GFP⁺ SLCs from different age groups were 167149/199912/262019 A.U. (2m/12m/24m), which got five times

biological repeats for each group and existed significant differences. Accordingly, we modified the Figure Legends of **Fig. 3a** in the manuscript. **The corresponding data was added as Fig. 3a-b.**

We sincerely thank for your comments and hope these revised figures will show clear presentation and interpretation of our data.

Point 5: The choice of Nestin as a cytoskeletal component undergoing changes is arbitrary and lacks a good rationale and controls. What happens to other intermediate filaments? The role of Nestin for ER-mitochondria contacts is not demonstrated with the data presented. Also, the authors do not investigate the functional significance of Nestin for ER-mitochondria contacts. This remains a mystery. Later in the manuscript, Nestin amounts are correlated with Leydig cell presence and proliferation. It remains unclear whether this is one of many factors that could lead to problems during ageing and even more, whether it is a key factor.

Response: We feel sorry for the unclear description due to our oversight during the preparation of the original manuscript. For the first concern, it is true that the intermediate filament is an enormous protein family with great heterogeneity in specific tissues or organs. For examples, Type III intermediate filament, Desmin, maintains structure and transmits tension signals in muscles; Type II intermediate filament, Keratin, is specifically expressed in epithelial cells and can resist tension and maintain epidermis integrity (*Cold Spring Harb Perspect Biol.* 2017 Dec;9(12):a021642; *Br J Dermatol.* 2020 Mar;182(3):636-647). Therefore, the choice of Type IV intermediate filament, Nestin, was not arbitrary by the reasons that on the one hand, in our previous studies, Nestin was already illustrated to be a specific marker of SLCs in testis (*Cell Res.* 2014 Dec;24(12):1466-85), while other intermediate filaments have not been reported to be expressed in stem Leydig cells. On the other hand, we have reported that Nestin could also influence mitochondrial function and regulated the maintenance and stemness in stem/progenitor cells (*Stem Cells.* 2018 Apr;36(4):589-601). Therefore, we hypothesized that Nestin, rather than other intermediate filaments, might play a

crucial role in regulating SLC differentiation. As suggested, in order to let the readers further understand and make the logic smoother, we added sentences as “Nestin, the major expression of intermediate filaments in SLCs” in the revised manuscript.

For your second concern, actually, as for we previously engaged in research about Nestin for a long time, we have already got two Nestin-short hairpin RNAs (shNES-1 or shNES-2) to specifically reduce Nestin expression in SLCs (*Nat Commun.* 2019 Nov;10(1):5043; *Nat Commun.* 2018 Sep;9(1):3613; *J Hepatol.* 2021 May;74(5):1176-1187; *Eur Respir J.* 2021 Oct;2003721). Also, we have already performed a series of Nestin-knockdown experiments to illustrate the functional significance of Nestin in mito-ER contact. However, due to the space limitations, we did not provide them in the original manuscript. Details were as follows:

First, we found that after knocking down Nestin, the level of mito-ER contacts proximity was reduced. Secondly, we investigated the functional significance of Nestin for mito-ER contact. The results showed that SLCs underwent senescence as evidenced by SA- β -Gal staining, along with increased intracellular ROS levels and decreased differentiation capacity. Altogether, These results might suggest that Nestin might be a key factor leading to dysfunctional mito-ER contact, thus impairing SLC intracellular homeostasis and differentiation capacity into LCs during ageing. **The corresponding data was now added as Extended Data Fig. 6.**

Point 6: Similarly, melatonin is one of very many antioxidants that could undergo changes during ageing. Why was this chosen over others? What happens to proteins of the thioredoxin family, known regulators of ER-mitochondria contacts (e.g., ERdj5, TMX1)? This is a critical flaw, since the authors are not able to rescue ER stress, as shown by the persistent elevated levels of BiP, suggesting other mechanisms are at play.

Response: We appreciated your inspiring comments. In fact, in previous studies, we have revealed that Keap1 could bind more Nestin and degrade it with SLC ageing in vitro, and that less Nestin could lead to the decrease of mito-ER contact. As some literatures have reported that the cellular redox levels could alter the ability of Keap1

to degrade proteins (*Cancer Cell*. 2017 Nov;32(5):539-541; *Physiol Rev*. 2018 Jul;98(3):1169-1203; *Mol Cell Biol*. 2003 Nov;23(22):8137-51; *Cancer Cell*. 2017 Nov;32(5):561-573.e6), the main purpose of this in vivo experiment was to use antioxidants to influence Nestin degradation and thus improve mito-ER contact. Actually, we've already used other classical antioxidants, such as NAC and VitE, which were broadly used in experiments (*Cell*. 2019 Jul;178(2):330-345.e22; *Nat Immunol*. 2020 Jul;21(7):727-735). Due to the **space limitations on the amount of data in the article**, we **only provided the data of melatonin treatment** in the original manuscript, mainly because of its excellent anti-oxidation property (*J Pineal Res*. 2015 Oct;59(3):365-75; *J Pineal Res*. 2016 Oct;61(3):253-78) and its safety in clinic (*BMJ*. 2006 Feb;332(7538):385-93; *Nutrients*. 2020 Aug;12(9):2561). Data from animal and human studies have proved that short-term use of melatonin only elicits mild adverse effects (*Clin Drug Investig*. 2016 Mar;36(3):169-75), and oral melatonin supplements are already approved by FDA and are accessible in normal pharmacies. Therefore, we believe that it will be of greater significance if the effects of melatonin in vivo in attenuating male reproductive ageing can be further illustrated. Of course, other drugs that alter redox levels could also rescue mito-ER contact deficiency and restore the differentiation capacity in aged SLCs. As suggested, we provided data on other antioxidant treatments in the revised manuscript, and we hope will clarify our therapeutic mechanisms: Antioxidants could react upon elevated ROS stress through Nrf2/Nestin-Keap1 binding pattern to tether the mito-ER contact and improve the differentiation capacity in aged SLCs. **The corresponding data was added as Extended Data Fig. 10.**

For the second concern, the induction and relief of ER stress indeed are associated with numerous ER/mitochondria membrane-bound proteins. TMX1 actively mediates ER state through controlling mito-ER Ca²⁺ cross talk (*J Cell Biol*. 2016 Aug 15;214(4):433-44). ERdj5 recognizes and unfolds misfolded proteins for their efficient retrotranslocation (*Science*. 2008 Jul;321(5888):569-72). However, although melatonin could not be able to rescue ER stress completely, there still was a partial

remission after melatonin treatment, which got a statistical significance, indicating its feasibility to apply antioxidants to alleviating the degeneration of aged SLCs. In addition, as suggested, we measured the exact changes of the thioredoxin family protein (e.g., ERdj5, TMX1) after melatonin treatment. The results showed that the expression levels of ERdj5 and TMX1 were both elevated after melatonin treatment, further indicating that melatonin plays a vital role in rescuing ER stress. **The corresponding data was added as Extended Data Fig. 8f-h.**

Point 7: There are no functional links between melatonin and ER-mitochondria contacts in the manuscript other than a non-quantified fluorescence microscopy in Figure 5d. Of course, a clear association of melatonin availability with ER-mitochondria contact formation would require massive amounts of work.

Response: We thank the reviewer for bringing this point to our attention. Actually, we have already done large amounts of work and identified Nestin as a functional linker/an important regulator between melatonin and mito-ER contact. However, considering the space limitations, we mainly emphasized the final effects of melatonin treatment in male reproductive system ageing, thus we did not provide part of the work about the intermediate link (Nestin) between melatonin and mito-ER contact in the original manuscript. As suggested, we reorganized the corresponding data and now the main content included the following aspects:

First, we mechanically verified that melatonin could react upon elevated ROS stress through promoting Nrf2 competitively binding to Keap1, thus releasing Nestin to prevent it from proteasome degradation. The corresponding data was added as Fig. 5a-c.

Secondly, we confirmed that the mito-ER contact got closer after melatonin treatment. The fluorescence microscopy and additional transmission electron microscope analysis both showed that melatonin treatment in vivo could restore mito-ER contact in SLCs from aged mice, providing functional link between melatonin and mito-ER contact. **The corresponding data was added as Fig.5d and Fig. 6c-e.**

Furthermore, in order to **highlight the functional link between melatonin and mito-ER contact**, we have done the loss-of-function experiments by using the Nestin-short hairpin RNA (shNES-2) to specifically reduce Nestin expression in SLCs in vitro (*Nat Commun.* 2019 Nov;**10(1):5043**; *Nat Commun.* 2018 Sep;**9(1):3613**; *J Hepatol.* 2021 May;**74(5):1176-1187**; *Eur Respir J.* 2021 Oct;**2003721**) and the results showed that the less mito-ER contact with disordered redox homeostasis and reduced capacity of SLC differentiation were both failed to be restored by melatonin treatment in SLCs after knockdown of Nestin, which were consistent with the observations in vivo (**Fig. 7 and Extended Data Fig. 14**). **The corresponding data was added as Extended Data Fig. 12.**

Altogether, these above data suggest that there exists a clear association of melatonin availability with mito-ER contact formation through Nestin.

Reviewer #3:

Point 1: Mitochondrial Morphology: could the author provide the average value which was measured in nanometers, and not as % of 2m (or, could they add the actual values at least in the supplementary materials?)

Response: Thank you for your advice. As suggested, we examined the average value of mitochondrial lengths measured in nanometers according to the previous protocols (*Science*. 2013 Jan;339(6118):464-7). The results showed that mitochondria underwent prominent morphological change during SLC senescence. **The corresponding data was added as Fig. 3h and i.**

Point 2: What is the average mito-ER contact sites width in the different conditions (nanometers)? What about the length? These parameters can be determined by TEM images analysis.

Response: We are grateful for your kind advice. As suggested, we performed TEM analysis, both in vitro and in vivo, and have examined average of mito-ER contact sites width in different conditions measured in nanometers according to the previous protocols (*Elife*. 2018 Apr;7:e32866; *Cell Metab*. 2020 Apr;31(4):791-808.e8). Moreover, we have already provided corresponding in vivo data, such as “mito-ER proximity (nm)” (**Fig. 7c**) in our original manuscript. As suggested, we further re-measured the average of mito-ER contact sites width (nanometers) in TEM images analysis of **Figure 2e** and the results confirmed that lower mito-ER contact sites width could be observed in aged SLCs in vitro. **The corresponding data was added as Fig. 2e-g and Fig. 7b-c.**

Point 3: It is surprising to me that PLA works with a protein residing in the lumen of ER shows a positive signal (normally for contact sites, proteins residing on the surface of the 2 organelles are used); is there any publication in which calreticulin has been described at MAMs? PLA is used for detection of protein-protein interactions at

distances <40 nm. Could the author address this point?

Response: Thank you for your inspiring query. Actually, although calreticulin was long considered to be a Ca²⁺ binding protein that locates inside the ER lumen, it was proved that the distribution of calreticulin was also enriched in mitochondrial-associated ER membrane (MAM) (*Cell*. 2007 Nov;131(3):596-610). Moreover, the activity and action of calreticulin are regulated by other chaperones or proteins most likely occurring at the MAM of the ER (*Trends Cell Biol*. 2009 Feb;19(2):81-8), which strengthens the idea that calreticulin can be found in MAM, a membranous ER structure that is in proximity with mitochondria. These works together may explain why calreticulin, a protein that resides in the lumen of ER, can be detected to be spatially close to mitochondrial membrane-bound protein.

Furthermore, it would be a better idea to use a classical protein complex, which makes ER bridge to mitochondria. As suggested, we used inositol 1,4,5-triphosphate receptors (IP3R1) as a marker for ER and Glucose-regulated protein 75 (GRP75) as a mitochondrial marker to represent the proximity of these two organelles according to the previous protocols (*Proc Natl Acad Sci U S A*. 2019 Dec 10;116(50):25322-25328; *MBO Rep*. 2019 Sep;20(9):e47928). Accordingly, the results showed that SLCs establish less mito-ER contact during ageing. **The corresponding data was added as Fig. 2h-j.**

We sincerely appreciate your comments and hope the additional evidence will make our work more convincing.

Point 4: I have several concerns related to the TMRM analysis. First, was the dye used in quenching mode or not? 200nM seems a high concentration. According to the methods section, no Cyclosporin H was used. Controls (as FCCP, or oligomycin, or both) are missing.

Response: We appreciate the helpful comments you put forward. For the first concern, we are sorry for the unclear description about TMRE (tetramethylrhodamine, ethyl ester) analysis, which leads you to understand it as TMRM analysis. Actually, we used the

TMRE in non-quenching mode and we have done relevant concentration gradient experiments through research in order to determine the suitable concentration of TMRE for SLCs. On the other hand, we speculate that the different types of cells might result in the contradiction. For platelets, 10nM of TMRE is suggested (*Redox Biol.* **2019 Sep;26:101250**). Cardiac myocytes, HeLa cells and osteosarcoma cells are suggested to be stained at a concentration of 100nM (*Circ Res.* **2003 Oct;93(8):697-9**; *Autophagy.* **2020 Sep;16(9):1598-1617**; *Cell Mol Life Sci.* **2017 Apr;74(7):1347-1363**). Fibroblasts are suggested to be stained at a concentration of 200nM (*Cell Metab.* **2017 Dec;26(6):872-883.e5**). There are also other cell types that have been reported to be stained at a higher TMRE concentration, such as N2a cells (300nM) (*Autophagy.* **2019 Oct;15(10):1810-1828**) and CD4⁺ T cells (400nM) (*Cell Metab.* **2020 Jul;32(1):44-55.e6**). As a result, we found that 200nM was considered to be a suitable concentration and decided to use TMRE at 200nM when conducting the experiment.

As for the lack of use of Cyclosporin H, we learned that Cyclosporin H is a multidrug resistance inhibitor that allows a better loading of mitochondrial dyes in analysis and is often taken into consideration when the equilibrium is hard to reach due to the efflux of the dye (*Biotechniques.* **2011 Feb;50(2):98-115**). In our experiment, actually, we chose TMRE as the dye, which equilibrates quickly and exhibits a larger degree of binding to mitochondria than TMRM (tetramethylrhodamine, methyl ester) as well as other dyes (*Biophys J.* **1999 Jan;76(1 Pt 1):469-77**). We speculated that Cyclosporin H was often used in conjunction with TMRM rather than TMRE, possibly because of their different binding characteristics (*Blood.* **2007 Jun;109(11):4988-94**; *J Biol Chem.* **2011 Dec;286(48):41163-41170**). What's more, we didn't observe unreached equilibrium in our experiments, so we didn't take Cyclosporin H into account in the original manuscript.

For the second concern, actually, we have already set the positive control group such as FCCP (*Cell Metab.* **2020 Jul;32(1):44-55.e6**; *Proc Natl Acad Sci U S A.* **2018 Nov;115(48):12118-12123**), and have performed related experiments. However, due to the space limitations, we did not provide these sets of data in the original manuscript.

As the results shown in the revised manuscript, there existed significant differences in FCCP (positive control) group with 2m (experimental) group, indicating that the TMRE had a positive effect. **The corresponding description for TMRE analysis was further detailed in Method Section and the data was added as Extended Data Fig. 3g-h and Extended Data Fig. 7h-i.**

Point 5: DHE measurement: a positive control (for example, antimycin) is missing.

Response: Thank you for your comments. As the same with the previous question, we have done all controls in different experiments but have not presented them in the original manuscript. Actually, antimycin have already been used as a positive control in detecting intracellular ROS levels according to previous protocols (**Asian J Androl. Sep-Oct 2020;22(5):465-471; *Oncotarget.* 2017 Dec;9(3):3459-3482**). According to the suggestions of reviewers, we added controls to better present the reliability of data for ROS analysis. **The corresponding description was further detailed in Method Section and the data was added as Extended Data Fig. 3c-d.**

Point 6: Figure 3 is entitled: “Less mito-ER contact results in mitochondrial dysfunction and ER stress in aged SLCs”. However, the authors do not provide a formal demonstration of this point. At variance, figure 3 provide evidence that conditions associated with lower mitoER contact sites (that is, 12 and 24 months) are also characterised by changes in the expression levels of several ER stress-related genes and anti-oxidative genes. To really demonstrate that less contact sites results in mito dysfunction in SLCs they should: (i) characterise which MAMs resident protein is downregulated during ageing; (ii) change the expression of MAMs resident proteins and check the effects on ER stress and REDOX genes.

Response: Thank you for your professional advice. MAMs resident protein plays an important role in mitochondrial dysfunction and ER stress (***Cell.* 2007 Nov 2;131(3):596-610, *Curr Top Microbiol Immunol.* 2018;414:73-102**). As suggested, we examined five major MAMs resident proteins (IP3R2, MFN2, PACS2, GRP75 and

Sig1R) during SLC ageing and surprisingly found that the expression of MAMs resident proteins were all increased, though the contact between ER and mitochondria was lessened, which was consistent with other aged cells (*Cell Death Dis.* 2018 Feb;9(3):332; *Cell Rep.* 2020 Sep;32(10):108125; *Crit Rev Biochem Mol Biol.* 2019 Dec;54(6):517-536). Because the cytoskeleton often works as scaffolding, the ER and mitochondria might be unanchored when the level of Nestin decreased, which could result in lessened area of MAMs. Together, we speculated the reduction of mitochondria-ER contact due to the change of Nestin might trigger a compensatory up-regulation of MAMs resident proteins.

Furthermore, we downregulated the expression of the five major MAMs resident proteins by RNA interference as suggested. The results showed the reduced levels of antioxidation genes and elevated levels of ER stress genes in SLCs when the MAMs resident proteins were knockdown. As a result, these data confirmed that MAM resident proteins were indeed crucial to play important biological functions and needed to maintain at a certain level in cells. However, it probably was passively altered, which indicated that it might be the downstream of Nestin-dependent mito-ER contact.

We sincerely appreciate your comments and hope the additional evidence will provide a more formal demonstration of this point. **The corresponding data was presented as Attached Fig. 1.**

Attached Fig. 1 | The change of expression of MAMs resident proteins and their effects on REDOX genes and ER stress.

(a-e) qPCR analysis of relative mRNA expression of IP3R2, Western Blot analysis and quantification of IP3R2 protein expression and qPCR analysis of relative mRNA expression of anti-oxidative and ER stress related genes of IP3R2-knockdown primary SLCs from 2 months old mice. (n = 3 biological repeats for each group; All data are mean ± SD; Multiple unpaired t tests).

(f-j) qPCR analysis of relative mRNA expression of MFN2, Western Blot analysis and quantification of MFN2 protein expression and qPCR analysis of relative mRNA expression of anti-oxidative and ER stress related genes of MFN2-knockdown primary SLCs from 2 months old mice. (n = 3 biological repeats for each group; All data are mean ± SD; Multiple unpaired t tests).

(k-o) qPCR analysis of relative mRNA expression of PACS2, Western Blot analysis and quantification of PACS2 protein expression and qPCR analysis of relative mRNA

expression of anti-oxidative and ER stress related genes of PACS2-knockdown primary SLCs from 2 months old mice. (n = 3 biological repeats for each group; All data are mean \pm SD; Multiple unpaired t tests).

(p-t) qPCR analysis of relative mRNA expression of GRP75, Western Blot analysis and quantification of GRP75 protein expression and qPCR analysis of relative mRNA expression of anti-oxidative and ER stress related genes of GRP75-knockdown primary SLCs from 2 months old mice. (n = 3 biological repeats for each group; All data are mean \pm SD; Multiple unpaired t tests).

(u-y) qPCR analysis of relative mRNA expression of Sig1R, Western Blot analysis and quantification of Sig1R protein expression and qPCR analysis of relative mRNA expression of anti-oxidative and ER stress related genes of Sig1R -knockdown primary SLCs from 2 months old mice. (n = 3 biological repeats for each group; All data are mean \pm SD; Multiple unpaired t tests).

Point 7: Figure 4b. Immunofluorescence is a semi-quantitative technique; the authors should confirm cytoskeletal proteins expression by Western Blot.

Response: We are sorry for the unclear description in the original manuscript. Actually, the results of Western Blot and its quantitative analysis have already been provided as **Fig. 4g and h** in the original manuscript. The results showed that intermediate filament Nestin expression sharply decreased in aged SLCs, yet little change was present in microtubule, α -tubulin, and microfilament, F-actin. **The corresponding data was presented as Fig. 4g and h.**

Point 8: Figure 4f: the term “nestin organization” needs to be defined better. What did the authors quantify exactly?

Response: We are sorry for the unclear description for heterogeneity of Nestin in **Fig. 4c and f** in the original manuscript. As the **Figure 4c** shown, in young SLCs, Nestin has a uniform organization in cytoplasm; whereas in old SLCs, Nestin has an irregular and uneven organization in cytoplasm, indicating that the differences of Nestin staining

intensity markedly exist. Zhang et al. inspiringly evaluated heterochromatin organization heterogeneity by calculating pixel-to-pixel coefficient of variation value of Hoechst 33342 with ImageJ (*Science*. 2015 Jun;348(6239):1160-3). Given this, following the same method, we evaluated the organization of Nestin by the formula: coefficient of variation=Standard deviation/Mean in order to represent the organization heterogeneity of Nestin. Altogether, the result showed that intermediate filament Nestin expression sharply decreased with a more uneven distribution. **The corresponding description was detailed in Figure legend Section of Fig. 4f.**

Point 9: With reference to figure 6: what happens in this model (e.g. 24 months+mel, with respect to 24 months only) to contact sites, after treatment with Melatonin?

Response: We appreciate the reviewer's concerns regarding the change of mito-ER contact sites after treatment with melatonin in vivo. Actually, we have detected that mito-ER contact sites were increased after melatonin treatment in vivo through fluorescence microscopy in **Fig. 5d** in the original manuscript. In additional, as suggested, we also performed additional experiments to investigate the change of mito-ER contact sites in SLCs from 24 months old mice after treatment with melatonin in vivo. The transmission electron microscope analysis results showed that melatonin treatment in vivo could restore mito-ER contact in aged SLCs, providing stronger proof for the relationship between Nestin and mito-ER contact. **The corresponding data was presented as Fig. 6c-e.**

Point 10: The parameters that authors take into account to study mito ER contacts should be consistent (they use percentages in figure 2, and proximity in nm in figure7). Which contact sites range do the author consider in the different models proposed (I refer to Figure 2 and 7): do they always analyse contact sites <100nm? If not, why?

Response: We thank the reviewer for bringing this point to our attention. For the first concern, in order to unify the statistical standard for mito-ER contact as suggested, we have reanalyzed relative amount of mito-ER apposition (% covered by ER) and the

mito-ER proximity both in **Figure 2 and 7** in the original manuscript, and the result showed that less effective apposition sites could be observed in SLCs from older mice, consistent with increased mito-ER contact distances (**Fig. 2e-g**) and by conditionally knockout of Nestin expression in PDGFR α ⁺ SLCs in testis, ER and mitochondria exhibited significant detachment compared to control group, but which was failed to be rescued by melatonin treatment (**Fig. 7b-c**). **The corresponding data was presented as Fig. 2e-g and Fig. 7b-c.**

For the second concern, in the present study, we measured the mito-ER distances according to the methods in the following articles in vitro (*Elife*. 2018 Apr;7:e32866) and in vivo (*Cell Metab*. 2020 Apr;31(4):791-808.e8). Actually, the physical tethers between mitochondria and the endoplasmic reticulum measured by Csordas et al. range from 9 to 30 nm (*J Cell Biol*. 2006 Sep;174(7):915-21). However, the morphological alteration of ER and mitochondria in different conditions could affect the average distance between mitochondria and the endoplasmic reticulum (*Proc Natl Acad Sci U S A*. 2016 Oct;113(40):11249-11254). Taking this into account, we repeated the quantification of the mito-ER distances in **Figure 2** using a variety of interorganellar tethering (mito-ER contacts) distances (e.g., <25 and <50 or <100 nm), and this data was shown in **Extended Data Fig. 2c-f**. We found that, irrespective of the chosen distance, the effect of senescence on the contact between mitochondria and the endoplasmic reticulum in SLCs is the same. Therefore, we next chose one consistent interorganellar distance to quantify mito-ER contact for the remainder of our study both in vitro and in vivo. **The corresponding data was presented as Extended Data Fig. 2c-f.**

Point 11: Extended data figure 2: control are missing (example, FCCP; additional western blots to study autophagy related protein expression, such as for example LC3 I and II; etc.)

Response: For the first concern, actually, the experiments using CellROX, DHE and TMRE staining have already been conducted with positive control groups according to

the previous protocols (*Front Pharmacol.* 2017 Sep;8:648; *Asian J Androl.* Sep-Oct 2020;22(5):465-471; *Oncotarget.* 2017 Dec;9(3):3459-3482; *Cell Metab.* 2020 Jul;32(1):44-55.e6; *Proc Natl Acad Sci U S A.* 2018 Nov;115(48):12118-12123). However, due to the space limitations, we did not provide these sets of data in the original manuscript. According to the suggestions of reviewers, we added these controls to better present the reliability of data. **The corresponding description for ROS analysis and TMRE analysis was further detailed in Figure Legend and Method Section and the data was added as Extended Data Fig. 3a-d, Extended Data Fig. 3g-h and Extended Data Fig. 7h-i.**

For the second concern, as suggested, we performed additional western blots to study autophagy related protein expression, such as p62, LC3 I and II (*Cell Metab.* 2020 Jul;32(1):44-55.e6). The results showed LC3-II expression decreased and p62 accumulated as SLCs aged, which indicated the global level of autophagy decreases, while the local level of autophagy (mitophagy) may increase. **The corresponding data was presented as Extended Data Fig. 3k-m.**

Point 12: Extended figure 5: what happens in case of treatment with melatonin of 2m old SLCs (e.g. without H2O2)?

Response: Actually, we have already set the group (2m old SLCs+melatonin), and conducted experiments parallel to the other 3 groups (2m old SLCs, 2m old SLCs+H₂O₂, 2m old SLCs+H₂O₂+melatonin). However, due to the space limitations and our topic focusing on the melatonin attenuating male reproductive ageing, we did not provide these sets of data in the original manuscript. As suggested, we added this group in the revised manuscript. **The corresponding data was presented as Extended Data Fig. 11.**

Point 13: Panel o of Extended data figure 5 needs to be incorporated in extended data figure 6.

Response: We apologize for our unclear description for **Extended Data Fig. 5 and 6** in the original manuscript. Actually, all the experiments in **Extended Data Fig. 5** are

conducted in vitro, while those in **Extended Data Fig. 6** are all in vivo experiments. Panel o of **Extended Data Fig. 5** measures testosterone concentration in the medium of Nestin-GFP+ SLCs. Therefore, we think it's inappropriate to incorporate it into Extended Data Fig. 6. We sincerely request the reviewers to allow us not to make adjustments. **The corresponding figure legends of present Extended Data Fig. 10 were revised to be more clarified.**

Point 14: Not sure References are listed correctly thorough the text. Please check.

Response: Thank you for your helpful advice. According to the reviewer's suggestion, we went through all the manuscripts and checked the **References** section, including reference consistency, number and format. We sincerely thank for your comments.

Point 15: Figure 2: what is the Total apposition length? Is this the mean surface covered by contacts on top of mito perimeter? Please specify better.

Response: We apologize for our unclear description of mito-ER contact analysis in Figure 2 in the original manuscript. Actually, we quantified the mito-ER contact according to the method from Gian-Luca McLelland's et al. works (*Elife*. **2018 Apr;7:e32866**). Briefly, the interaction length (referring to the interface length of ER tubules within 100 nm of the OMM) was quantified, as this distance is enough to capture tubules closely associated with the OMM (*J Cell Biol*. **2006 Sep;174(7):915-21**). "The total amount of ER-OMM apposition per mitochondria" exactly means the interaction length of ER tubules within 100 nm of the OMM covered by contacts per mitochondria. **The corresponding description for analysis was further detailed in corresponding Figure legend and Method Section, and the data was presented as Extended Data Fig. 2a.**

Reviewers' comments:

Reviewer #1 (Remarks to the Author):

I think the revised manuscript addressed my comments except Point1 (" overall, the letters in Figures are too small"). I cannot see any improvement in the font size, and I believe all of the letters should be enlarged.

Reviewer #2 (Remarks to the Author):

Yao et al. have revised their manuscript, where they propose that ER-mitochondria contacts are a key structure that promotes the survival and functionality of Leydig cells, responsible for the maintenance of the male reproductive system. They have successfully distinguished their observation from previous research on actin. The investigation of redox-related effects on MERCs, including the reductant TMX1 and of antioxidants (e.g., Extended Data 10) appears to suggest that hyperoxidation of MERC proteins is behind the observations. This is, however, currently underdeveloped. The authors also do not provide a rationale for downregulating Nestin in aged animals. Since Nestin is already low in these, I do not understand why its knockout would make things better. Overall, the manuscript is preliminary, because it does not appropriately investigate autophagy and the MERC defect, where conflicting results are presented. In its current form, the manuscript cannot be published.

Specific points

1. The authors fail to use common nomenclature for contact sites between ER and mitochondria. These are usually called mitochondria-ER contacts (MERCs). Their biochemical isolate is called mitochondria-associated membranes (MAMs).
2. The PLA assay must be controlled. Are the IP3R and Grp75 signals as expected? IF for the antibodies used must be provided.
3. In the figure legend of 3a, it is stated that all Nestin+ cells are considered ROS-positive. This cannot be the case or is at a minimum a confusing statement. In that sense, the figure is not much improved over the previous version. Where are the ORS coming from? Tools exist to measure mitochondrial ROS. Therefore, this experiment is superficial. Moreover, the characterization of antioxidant proteins is very superficial. Just a few of many concerns: What happens to glutathione? What happens to thioredoxin?
4. The text should describe WHAT the mitochondrial morphology changes are rather than mention that changes occurred. It looks like mitochondria became smaller but that is unclear. Are the amounts of mitochondrial proteins different? This is a critical piece of information that is currently missing.
5. The autophagy characterization is inaccurate. At a minimum, the authors must block its progression and show how that influences key autophagy markers. Also, on the one hand, the authors claim that mitophagy is increased without providing any key characteristics but then go on that overall autophagy is decreased. This statement has no experimental basis with the current set of data and the authors must follow current guidelines for the analysis of autophagy.
6. The metabolic data in Extended Data 3n-p suggest that there is a decline in glycolysis, not of mitochondrial ATP. This is not accurately reported in the text. Moreover, GAPDH is frequently used as

a loading control. But this is inappropriate if glycolysis is altered.

7. The increased expression of key MERC promoters like Mfn2 and PACS2 suggests more, not less MERCs. The authors propose this observation indicates adaptation without following up on any of them. In that context, data presented in Extended Data 8 contradicts this hypothesis, since TMX1 promotes MERCs and is downregulated at 24 months.

8. The authors do not show a proper Nestin degradation analysis. On the one hand, they claim it is ubiquitinated but in the total Nestin lysate its amounts are unchanged. Apparently, this is because they only provide MG132-treated lysates. This must be done properly with the controls lacking proteasome inhibitors.

9. I am confused about the description of Figure 7, where the authors state that Nestin downregulation fails to attenuate reproductive aging. Is this not the expected outcome, given Nestin decreases during aging?

Reviewer #3 (Remarks to the Author):

Thanks for addressing all my concerns. I still have some concerns on the PLA based on Calreticulin and the OMM protein, as well as on GRP75 and IP3R. GRP75 is a protein sandwiched among VDAC and IP3R and honestly I am not sure it could be used as an OMM marker. PLA is a difficult technique, which relies on good antibodies, besides specific protein choice.

At this point, I suggest to remove this dataset that for me is not convincing.

Rather, Attached Fig. 1 | The change of expression of MAMs resident proteins and their effects on REDOX genes and ER stress. should be included in the manuscript as it enstrengthen the overall findings.

Response to decision letter

Reviewer #1 (Remarks to the Author):

Point 1: I think the revised manuscript addressed my comments except Point1 (" overall, the letters in Figures are too small"). I cannot see any improvement in the font size, and I believe all of the letters should be enlarged.

Response: Thank you for your advice regarding the font size. Accordingly, we have enlarged all the letters in the figures from **Fig.1 to Fig.7 and Extended Data Fig. 1 to Extended Data Fig. 17** in the revised manuscript and hope it will show a clearer presentation.

Reviewer #2 (Remarks to the Author):

Point 1: The authors fail to use common nomenclature for contact sites between ER and mitochondria. These are usually called mitochondria-ER contacts (MERCs). Their biochemical isolate is called mitochondria-associated membranes (MAMs).

Response: Thank you for the helpful suggestions. As suggested, we have revised our manuscript and replaced expressions like “mito-ER contact” with the common nomenclature mitochondria-ER contacts (MERCs).

Point 2: The PLA assay must be controlled. Are the IP3R and Grp75 signals as expected? IF for the antibodies used must be provided.

Response: Thank you for your advice. As suggested, we have conducted IF as a control for the antibodies used in our PLA assay and found that the IP3R1 and Grp75 signals were as expected, supporting our conclusion of our PLA assay. Additionally, in order to further confirm our conclusion, we performed additional immunofluorescent staining and conducted surface-surface contact site area assay of VDAC and IP3R1, which showed that SLCs establish less MERCs during ageing. **The corresponding data was presented as Fig. 2h-j and Extended Data Fig. 2g-j.**

Point 3: In the figure legend of 3a, it is stated that all Nestin+ cells are considered ROS-positive. This cannot be the case or is at a minimum a confusing statement. In that sense, the figure is not much improved over the previous version. Where are the ORS coming from? Tools exist to measure mitochondrial ROS. Therefore, this experiment is superficial. Moreover, the characterization of antioxidant proteins is very superficial. Just a few of many concerns: What happens to glutathione? What happens to thioredoxin?

Response: We appreciate the helpful comments you put forward. For the first concern, we apologize for our unclear statement in describing ROS level in SLCs. Actually, we meant to quantify ROS level in SLCs from different age groups through mean fluorescence intensity of DHE staining, instead of defining ROS-positive cell group.

(*Cell*. 2020;181(6):1307-1328.e15; *Circulation*. 2005;111(18):2347-55; *Cell*. 2015;163(3):560-9). Accordingly, we modified the **Figure Legends** of **Fig. 3a** in the manuscript. **The improved corresponding data was added as Fig. 3a and b.**

For the second concern, mitochondria are the main oxygen consuming organelles in cells, and historically have been considered as a principal intracellular source of reactive oxygen species (ROS). (*Cell Metab*. 2020 Mar 3;31(3):642-653.e6; *Cell Metab*. 2012 Apr 4;15(4):451-65). As suggested, we performed additional flow cytometry using MitoSOX staining to measure the mitochondrial ROS level in SLCs from different age groups in vivo. The result showed that the elevation in cellular ROS level mainly results from excessive ROS production in mitochondria. **The corresponding data was added as Extended Data Fig. 3a-f.**

Additionally, we fully appreciate your concerns regarding the characterization of antioxidant proteins, and as suggested, we have detected the change of glutathione and thioredoxin following the established protocol (*J Neurosci*. 2017 Jun;37(23):5770-5781). The results showed that the cytosolic glutathione and thioredoxin antioxidant defense systems are damaged during SLC ageing, which is consistent with the decrease of other antioxidant proteins we measured in our study (**Fig. 3c-e**). Above all, these results support our idea that aged SLCs exhibit disrupted redox homeostasis, manifested by ROS accumulation and damaged antioxidant defense systems. **The corresponding data was added as Extended Data Fig. 3g-m.**

Point 4: The text should describe WHAT the mitochondrial morphology changes are rather than mention that changes occurred. It looks like mitochondria became smaller but that is unclear. Are the amounts of mitochondrial proteins different? This is a critical piece of information that is currently missing.

Response: We feel sorry that we failed to clarify the exact changes of mitochondrial morphology in our manuscript. Actually, we have already measured the change of length and size of mitochondria during SLC ageing (**Fig. 3f-i**), and have stated that mitochondria become smaller in aged SLCs in the Figure Legend for **Fig. 3g** instead of in the Result section. **We sincerely thank for your comments and added detailed**

description for this morphological change of mitochondria in the Result section as suggested.

Additionally, we performed experiments to measure the amounts change of mitochondrial marker proteins, which represent mitochondrial mass, such as mitochondrial membrane proteins, TOM20 and TIM23. (*Redox Biol.* 2017 Apr;11:297-311; *Sci Rep.* 2017 Sep 6;7(1):10710). The results showed that the protein levels of TOM20 and TIM23 all decrease during SLC ageing, which is consistent with altered mitochondrial mass in other senescent cells (*EMBO J.* 2016 Apr;35(7):724-42; *Int J Mol Sci.* 2021 Aug;22(17):9171. *J Gerontol A Biol Sci Med Sci.* 2011 Nov;66(11):1178-85). **The corresponding data was added as Extended Data Fig. 4a and b.**

Point 5: The autophagy characterization is inaccurate. At a minimum, the authors must block its progression and show how that influences key autophagy markers. Also, on the one hand, the authors claim that mitophagy is increased without providing any key characteristics but then go on that overall autophagy is decreased. This statement has no experimental basis with the current set of data and the authors must follow current guidelines for the analysis of autophagy.

Response: Thank you for your professional advice. For the first concern, in order to confirm the autophagic response, we followed the current guidelines for the analysis of autophagy flux and performed an additional set of experiments using the vacuolar H⁺ ATPase inhibitor bafilomycin A1 to block its progression (*Autophagy.* 2019 Sep;15(9):1606-1619; *Autophagy.* 2021 May;17(5):1142-1156). Flux experiment demonstrated that the degradation of key autophagic markers p62 and LC3 were blocked by bafilomycin A1, indicating that overall autophagy level decreased during ageing and that melatonin treatment can upregulate overall autophagy level in aged SLCs. **The corresponding data was added as Extended Data Fig. 4g-i and Extended Data Fig. 8a-c.**

For the second concern, considering that PINK1/Parkin-driven mitophagy is the most characterized mitophagy pathway (*J Cell Biol.* 2020 Nov 2;219(11):e202004029;

J Pineal Res. 2016 May;60(4):383-93; *Redox Biol.* 2017 Apr;11:297-311), we measured the protein expression level of PINK1 and Parkin during SLC ageing. Results indicated that the upregulation of PINK1/Parkin pathway may account for the rise in mitophagy level, confirming that the mitophagy was enhanced during SLC ageing, which was also consistent with our IF data (*Autophagy.* 2015;11(3):547-59; *Autophagy.* 2021 Oct;17(10):3011-3029). **The corresponding data was added as Extended Data Fig. 4j-n and Extended Data Fig. 8d-h.**

Overall, we found that SLCs characterized with decreased overall autophagy but with enhanced local mitophagy during ageing. However, the correlation between autophagy and mitophagy is not the focus of our research in this paper, which could be conducted in the future studies to fully elucidate the mechanism of the imbalance of autophagy during SLC ageing.

Point 6: The metabolic data in Extended Data 3n-p suggest that there is a decline in glycolysis, not of mitochondrial ATP. This is not accurately reported in the text. Moreover, GAPDH is frequently used as a loading control. But this is inappropriate if glycolysis is altered.

Response: Thank you for bringing this point to our attention. For the first concern, we apologize for our unclear text description. Actually, the data showing the decreases in glucose uptake and lactate biosynthesis indicated changes in overall cellular metabolism, referring to a decline in glycolysis, instead of representing mitochondrial function. Only the ATP content is the indicator that represents mitochondrial function. As suggested, we revised the expression and made it more accurate, which was consistent with our data.

For the second concern, thank you for your professional advice. GAPDH is one of highly conserved key enzyme in glycolysis, but when the glycolytic capacity of cell/tissue is changed, GAPDH may experience different change under different conditions. In some cases, the expression of GAPDH was altered in ageing skeletal muscle, indicating it was not a proper internal control (*J Gerontol A Biol Sci Med Sci.* 2000 Mar;55(3):B160-4). However, in many other cases, there was a stable expression of

GAPDH when the changes happened in other glycolysis-related proteins such as phosphofructokinase (PFK); glucose transporter 1 (GLUT1); pyruvate kinase isozymes M2 (PKM2) or monocarboxylate transporter 4 (MCT4) (*Nature*. 2020 Feb;578(7796):621-626; *PLoS Pathog*. 2017 Jul;13(7):e1006503; *World J Gastroenterol*. 2016 Oct;22(38):8519-8527). In our study, to further confirm that GAPDH can be used as a loading control, we performed additional experiments to compare the expression of GAPDH with other reference proteins, such as β -actin, α -tubulin, and found that there was no significant difference in the expression of GAPDH in SLCs from different age groups. Thus, GAPDH here was reasonable to be used as an internal reference. **The corresponding data was presented as Attached Fig. 1.**

Attached Fig. 1 | The protein expression of GAPDH in SLCs from different age groups.

(a-c) Western Blot analysis and quantification of GAPDH protein expression normalized to different reference proteins. (n = 3 biological repeats for each group; All data are mean \pm SD; Multiple unpaired t tests).

Point 7: The increased expression of key MERC promoters like Mfn2 and PACS2 suggests more, not less MERCs. The authors propose this observation indicates adaptation without following up on any of them. In that context, data presented in Extended Data 8 contradicts this hypothesis, since TMX1 promotes MERCs and is downregulated at 24 months.

Response: We appreciate your professional comments and we feel sorry for the unclear statement for the relationship between the change of these mitochondria-associated ER membrane (MAM) proteins and the MERCs during SLC ageing. Actually, the

molecular composition of the MAM proteins is complicated and they play different roles in pivotal cellular processes during ageing (*Cell Death Dis.* 2018 Feb 28;9(3):332). Given this, the direct or indirect links between senescence and MAM proteins or MERCs were currently unknown. As a result, the expression of some MAM proteins may increase and others may decrease, which was summarized by the review (Attached Table. 1, *Commun Biol.* 2021 Nov;4(1):1323).

Function	Name	Effect of experiment on protein levels/activity (-/+)	Effect on MERCs number	Effects on ER and/or MT	Effects on senescence-associated phenotype	Ref.	
ER-MT TETHERING	MFN1	+ (Stabilization)	N/D	Increased MT mass	Induction of senescence	113	
	MFN2	- (KO)	N/D	N/D	Delayed RS	115	
	GRP75	+ (OE)	N/D	N/D	Delayed RS	118	
	FIS1	- (shRNA)	N/D	Hyperfused MT	Induction of senescence	157	
		+ (OE)	N/D	Rescue hyperfused MT	Rescue of DFO-induced senescence	156	
ER-MT METABOLITES TRANSFERS	ITPR1	- (shRNA)	N/D	N/D	Limited OIS	61	
	ITPR2	- (shRNA)	N/D	Limited ER-MT calcium fluxes/ROS/ MMD	Limited OIS and delayed RS	61	
		- (siRNA)	N/D	Limited ER-MT calcium fluxes/MT ROS	Reduction of senescence	62	
		- (KO)	Decreased	Limited ER-MT calcium fluxes/MT ROS/ MMD	Reduction of senescence	46	
	ITPR3	- (shRNA)	N/D	N/D	Reduction of OIS	61	
	MCU	- (shRNA)	N/D	Limited ER-MT calcium fluxes	Reduction of OIS	61	
	ORP5	- (siRNA)	N/D	Alteration of MT morphology and reduced OXPHOS	Induction of senescence	125	
	SIGNALLING PROTEINS	PACS-2	- (KO)	N/D	N/D	Resistance to p53-dependent CCA and NF- κ B programme	131,142
		p66Shc	- (KO)	N/D	N/D	Delayed RS	135
		CISD2	- (KO)	N/D	Enhanced MMD	Reduction of cell proliferation	141
RelA		- (shRNA)	N/D	N/D	Inhibition of SASP	24	
NLRP3		- (KO)	N/D	N/D	Reduction of age-dependent increase of p53 and p21	27	
mTOR		- (Rapa)	N/D	N/D	Inhibition of NF- κ B-dependent SASP	26	
PML		- (KO)	N/D	N/D	Resistance to OIS	138	
	- (KO)	N/D	Limited ITPR3 phosphorylation Reduced ER-MT calcium fluxes	N/D	139		

Summary of the studies reporting a potential role for MERC-associated proteins involved in tethering, ER-MT metabolites transfers and signalling in the regulation of features of cellular senescence. For each study, the effect of MERCs protein dysregulation (upregulation + or downregulation-) on MERCs number and also ER and mitochondria are indicated. If investigated, the effects on features of cellular senescence are reported. N/D not determined, KO knockout, OE overexpression, Rapa Rapamycin, ER endoplasmic reticulum, MT mitochondria, OXPHOS oxidative phosphorylation, MMD mitochondrial membrane depolarization, OIS oncogene-induced senescence, RS replicative senescence, CCA cell cycle arrest.

Attached Table. 1 | Regulation of senescence-associated features by MERC-associated proteins.

In our study, we found an upregulation expression of some MAM proteins during SLC ageing (Extended Data Fig. 5a-c). Considering the reduction of MERCs due to the degradation of Nestin, it might lead to a compensatory upregulation of MAM proteins, which was consistent with the conclusion that the change in MAM protein levels does not always correlated with the MERCs number, indicating other molecular mechanisms regulating the number of MERCs (*Circulation.* 2019 Apr 16;139(16):1913-1936; *EMBO Rep.* 2010 Nov;11(11):854-60; *J Biomed Sci.* 2017 Sep 16;24(1):74) or the other cellular distribution or biological function (*Proc Natl Acad Sci U S A.* 2010 Jul 27;107(30):13342-7; *Cell Rep.* 2014 Sep 11;8(5):1545-57). On the other hand, we further detect that TMX1, a redox-sensitive oxidoreductase that is enriched on the MAM, is downregulated in aged SLCs (Extended Data Fig. 9f-h). The low expression of TMX1 induces oxidative stress, leading to excessive ROS, which

is the same phenomenon as in other cells (*EMBO J.* 2019 Aug;38(15):e100871; *Biol Direct.* 2021 Nov;16(1):22). And as a result, the lower level of the thioredoxin-related transmembrane protein also may contribute to the reduction of MERCs (*J Cell Biol.* 2016 Aug;214(4):433-44).

Above all, we feel grateful for your inspiring suggestions, as they enlighten us to seek for deeper understanding of the relationship between ageing and MAM composition, function or dynamics in the future studies. **Therefore, we added these points to the Discussion section.**

Point 8: The authors do not show a proper Nestin degradation analysis. On the one hand, they claim it is ubiquitinated but in the total Nestin lysate its amounts are unchanged. Apparently, this is because they only provide MG132-treated lysates. This must be done properly with the controls lacking proteasome inhibitors.

Response: We are grateful for your kind advice. As suggested, we have performed additional Western blot analysis of Nestin degradation analysis, following established protocol (*Nat Commun.* 2019 Nov 6;10(1):5043). The result showed that, in the absence of proteasome inhibitors, the protein expression level of Nestin in the total cell lysate decreases significantly in aged SLCs. **The corresponding data was added as Fig. 4i and j.**

Point 9: I am confused about the description of Figure 7, where the authors state that Nestin downregulation fails to attenuate reproductive aging. Is this not the expected outcome, given Nestin decreases during aging?

Response: We feel fully sorry for the misleading title of Figure 7. In this section, we are not conducting Nestin-KO experiment to attenuate reproductive ageing. Instead, we are trying to illustrate that Melatonin fails to attenuate reproductive ageing in the absence of Nestin. Accordingly, we have revised the title into “AAV-mediated downregulation of Nestin separates MERCs and diminishes Melatonin’s effect in attenuating male reproductive ageing.”, which is closer to our intention. We feel really grateful to you for pointing out the ambiguous expression in our manuscript and hope

the expressions in our revised article will show clear presentation.

At last, we sincerely want to express our gratitude towards Reviewer 2 for his/her patience and conscientiousness when reviewing our manuscript. He/she has provided very professional and inspiring advice in the expression, experiment design and logic part of our manuscript. His/her suggestions are helpful and have deepened our understanding in the research of MERCs.

Reviewer #3 (Remarks to the Author):

Thanks for addressing all my concerns. I still have some concerns on the PLA based on Calreticulin and the OMM protein, as well as on GRP75 and IP3R. GRP75 is a protein sandwiched among VDAC and IP3R and honestly I am not sure it could be used as an OMM marker. PLA is a difficult technique, which relies on good antibodies, besides specific protein choice.

At this point, I suggest to remove this dataset that for me is not convincing.

Rather, Attached Fig. 1 | The change of expression of MAMs resident proteins and their effects on REDOX genes and ER stress. should be included in the manuscript as it enstrengthen the overall findings.

Response: Thank for your professional advice. For the first concern, in order to confirm the formation of VDAC-GRP75-IP3R complex at the contact site between mitochondria and ER, we performed additional immunofluorescent staining and conducted surface-surface contact site area assay of VDAC and IP3R1. The result also indicated that MERCs are decreased during SLC ageing, which was consistent with our PLA assay. Therefore, we believe that the PLA assay of GRP75 and IP3R1 can still represent the formation of VDAC-GRP75-IP3R complex indirectly (*Diabetes. 2014 Oct;63(10):3279-94*). Besides, there are other reviewers that are interested in the colocalization between GRP75 and IP3R1. Therefore, we decided to move this figure to **Extended Data Fig. 2g-j.** and replace it with the surface-surface contact site area assay between VDAC and IP3R1. **The corresponding data was presented as Fig. 2h-j.**

For the second concern, thank for your inspiring comments. As suggested, we agreed to add the figure “The change of expression of MAMs resident proteins and their effects on REDOX genes and ER stress.” to **Extended Data Fig. 5d-w.**

REVIEWERS' COMMENTS

Reviewer #2 (Remarks to the Author):

The authors have addressed all comments in a satisfactory way.

Reviewer 2:

The authors have addressed all comments in a satisfactory way.

Response: Thank you for your approval. We would like to express our gratitude towards all the professional advice from the reviewer, which greatly increased the rigor and significance of this study.